psychology

gender bias, *H*-index, science impact, success rate, women in science, women underrepresentation

**Author for correspondence:**
Julia Astegiano
e-mail: juastegiano@gmail.com

†These authors contributed equally to this work.

# Unravelling the gender productivity gap in science: a meta-analytical review

Julia Astegiano[1,2,†], Esther Sebastián-González[1,3,†] and Camila de Toledo Castanho[1,4,†]

[1]Departamento de Ecologia, Instituto de Biociências, Universidade de São Paulo, Rua do Matão 321, trav14, 05508-090 São Paulo, Brazil
[2]Grupo de Interacciones Ecológicas y Conservación, Instituto Multidisciplinario de Biología Vegetal (IMBIV), Facultad de Ciencias Exactas, Físicas y Naturales, Universidad Nacional de Córdoba, Consejo Nacional de Investigaciones Científicas y Técnicas, CC 495, X5000ZAA Córdoba, Argentina
[3]Departamento de Biología Aplicada, Universidad Miguel Hernández, Avda. Universidad s/n, 03202 Elche, Spain
[4]Departamento de Ciências Ambientais, Universidade Federal de São Paulo—UNIFESP, Rua São Nicolau 210, 09913-030 Diadema, Brazil

JA, 0000-0003-0583-7291; ES-G, 0000-0001-7229-1845; CTC, 0000-0002-0198-3200

Women underrepresentation in science has frequently been associated with women being less productive than men (i.e. the gender productivity gap), which may be explained by women having lower success rates, producing science of lower impact and/or suffering gender bias. By performing global meta-analyses, we show that there is a gender productivity gap mostly supported by a larger scientific production ascribed to men. However, women and men show similar success rates when the researchers' work is directly evaluated (i.e. publishing articles). Men's success rate is higher only in productivity proxies involving peer recognition (e.g. evaluation committees, academic positions). Men's articles showed a tendency to have higher global impact but only if studies include self-citations. We detected gender bias against women in research fields where women are underrepresented (i.e. those different from Psychology). Historical numerical unbalance, socio-psychological aspects and cultural factors may influence differences in success rate, science impact and gender bias. Thus, the maintenance of a women-unfriendly academic and non-academic environment may perpetuate the gender productivity gap. New policies to build a more egalitarian and heterogeneous scientific community and society are needed to close the gender gap in science.

# 1. Introduction

> One is not born, but rather becomes, woman [1]

Women have traditionally been, and continue to be, underrepresented in science. Even though the percentage of women in science varies across regions, only 28.4% of the research and development employees in the world are female [2]. Gender inequalities are especially flagrant at the latest stages of the academic career when the 'leak' of female scientists out of the academic world is much larger than that of men [3–5]. Globally, women represent 53% of bachelor's graduates, 43% of PhD graduates and 28% of researchers [2]. This underrepresentation of female scientists can affect the quality and competitiveness of research centres, as ideas from heterogeneous groups are more feasible, effective and innovative [2,6]. Moreover, women should have the same opportunities as men to develop and present their own imprinting in the scientific endeavour to contribute to society [2].

The factors leading to the underrepresentation of women in science (i.e. the gender gap) are still under debate. One of the main claims is that women are less productive than men (i.e. that there is a gender productivity gap), and thereby, they author fewer scientific papers, receive fewer grants and are hired less frequently than men [5]. However, these claims do not necessarily mean that male scientists outperform their female counterparts. Many interacting socio-psychological and cultural factors may underlie the gender productivity gap. Men as a group may submit more papers and apply for more grants or faculty positions because they are more productive in a per capita basis, because they outnumber women (i.e. the gender gap itself) or because they persist longer in the scientific career than women, none of which seem to be determined solely by merit [7,8]. Alternatively, science produced by men may be of higher quality, leading to more grants or faculty positions. Some studies have found that papers authored by male scientists are more cited than those of female scientists [9], but this is not always the case [7,9]. Finally, conscious or unconscious gender bias against women in science may negatively affect their productivity. For example, Moss-Racusin *et al.* [10] found through an experimental approach that scientists rated a female candidate for a technician position to be less competent and hireable than a male candidate with an identical academic background. Such gender bias against women contributes to the productivity gap because it implies that a woman scientist needs to outperform a man to be perceived and evaluated as similar.

Many researchers and international organizations have raised their voices about the importance of addressing the gender inequality problem in science [2,11–15]. Such concern with gender parity in science has led to increasing efforts to promote female entrance and persistence in academia, for instance, by applying specific faculty programmes or promoting double-blind peer-review to avoid gender bias in the evaluation processes [16,17]. However, deciding on the best strategies to achieve gender equality in science implies deep knowledge of the causes underlying gender productivity differences, which is hard mostly because of the lack of systematic, thoughtful quantitative review studies of such causes. How does productivity vary among male and female scientists? Are productivity differences explained by a different success rate or only by the number of trials of each gender? Do men produce higher impact science? Is there a gender bias against women in science that can be evidenced by experimental studies? In an attempt to answer these questions, we quantitatively reviewed 110 studies (figure 1) evaluating gender differences in scientific productivity and their likely causes. First, we investigated gender differences in productivity both globally and across different research fields, periods of time and proxies of productivity. Then, we compared the success rate and the impact of scientific outcomes between genders as likely explanations of productivity patterns. Finally, we quantitatively reviewed results from experimental studies evaluating gender bias in science across different research fields.

# 2. Methods

## 2.1. Data compilation

We conducted a survey of published studies exploring different aspects of the differences in academic productivity between male and female scientists. On 27 April 2017, we collected candidate articles by searching the ISI Web of Knowledge database using the following combination of terms: (gap OR bias*) AND (gender OR 'women in science') AND (productivity OR publication* OR citation* OR 'research performance' OR grant OR referee*). The overall search

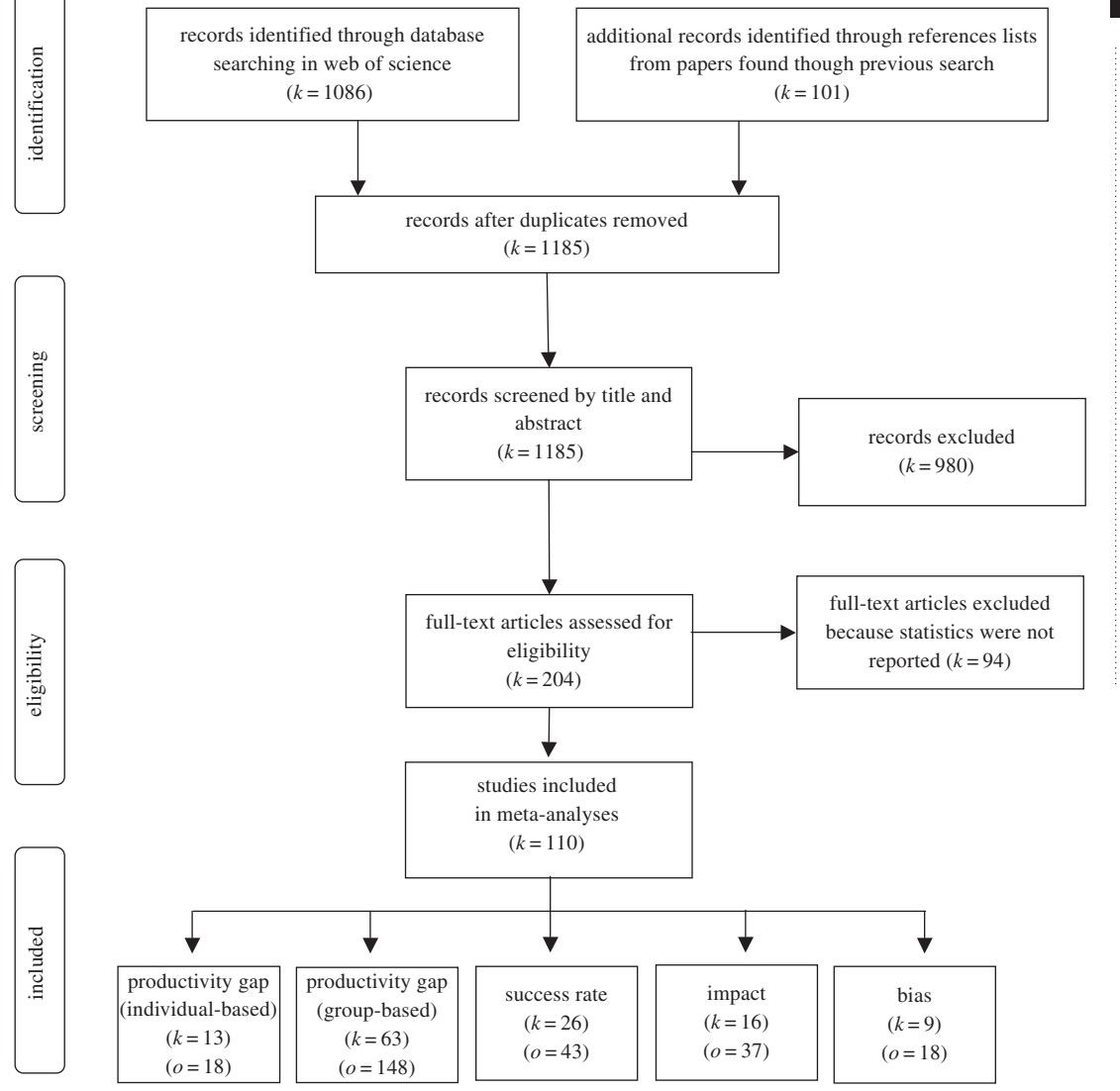

**Figure 1.** PRISMA flow diagram representing the flow of information through the decision process of searching and inclusion of articles and observations in the meta-analyses (modified from Moher *et al*. [18]). *k*, number of articles; *o*, number of observations.

led to 1086 candidate articles. Such articles were manually screened through title and summary to check that they included one of the four questions of our study (figure 1). The screening was done by the three authors and followed a conservative approach, so that only articles that were clearly out of the scope of the study were excluded at this point (e.g. articles comparing disease prevalence between males and females). We also scanned the bibliographies of the selected articles, which led to the addition of 101 new studies that were not originally detected in the first search in the ISI Web of Knowledge (figure 1).

The full text of the selected articles was reviewed to check their suitability for our meta-analyses following our eligibility criteria checklist (electronic supplementary material, table S1). First of all, we evaluated if the article quantitatively explored at least one of our four questions and provided the statistics required for the meta-analysis. Then, we assessed if the response variable evaluated, the study design and data type were included in our predefined list (see electronic supplementary material, table S2). When we found one study design, data or response variable that was not originally in our list but that we thought it could successfully address at least one of our questions, we personally discussed its inclusion. We also excluded opinion articles and studies that addressed our questions by means of reviews or surveys. Articles comparing wages between men and women were not considered, even if they were scientists. The three authors carefully extracted the required data from the text or tables in the articles (electronic supplementary material, datasets S1–S5). When the required data were only reported in graphical form, graphics were scanned and data were

royalsocietypublishing.org/journal/rsos　R. Soc. open sci. 6: 181566

extracted with DataThief III [19]. If an article explored one of our questions but lacked data, we contacted authors and asked for such data (number of articles ($k$) = 10).

When possible, articles reporting data from different journals were included as different observations, but only if they came from different research fields (electronic supplementary material, table S1). We separated observations when they came from different geographical regions (e.g. continent) or research positions (e.g. assistant and full professor). However, if an article evaluated different aspects of the same data (e.g. hierability and quality of a CV; women as first or last authors), we only included one of the proxies. When an article reported multiple observations across a given time period, each observation was considered different only when the time difference among observations was at least one decade. When such time difference was smaller, we only considered the most recent observation.

We created one dataset for each of the four study questions (electronic supplementary material, datasets S1–S5), except for the first question for which we created two datasets (electronic supplementary material, datasets S1 and S2). During the extraction process, we filled each dataset with the studies that directly addressed the specific question to be analysed. Some articles were suitable to investigate more than one question; thus, they were included in more than one dataset.

## 2.2. Q1: differences in scientific productivity

Two kinds of data addressed our first research question, i.e. how scientific productivity varies among male and female scientists: (i) individual-based studies comparing the average productivity of men and women (i.e. the sample unit was the researcher), and (ii) group-based studies exploring the proportion of a given productivity proxy (e.g. articles published in a given journal) attributed to men and women (i.e. the sample unit was the article). We decided to analyse them separately because the patterns emerging from these two different kinds of data may be explained by different mechanisms (electronic supplementary material, datasets S1 and S2).

We used Hedges' $d$ [20] to measure the effect size of gender on researcher's productivity in individual-based studies. To calculate such effect size, we used the mean, standard deviation and sample size information for male and female researchers for the corresponding productivity proxy in each primary study. Positive Hedges' $d$ effect sizes indicate higher male productivity, whereas negative ones indicate higher female productivity. The effect size of gender on researcher's productivity in group-based studies was measured by calculating the raw proportion. Such raw proportion was calculated as the number of productivity outputs attributed to men divided by the total number of productivity outputs [21]. Raw proportion values higher than 0.5 indicate higher male productivity, whereas those smaller than 0.5 indicate higher female productivity.

We performed hierarchical mixed-effects meta-analyses, which considered the lack of independence among the effect sizes of observations obtained from the same article. Both an article-level and an observation-level random effect were included as nesting factors to incorporate such lack of independence. Heterogeneity among effect sizes within each dataset was assessed by calculating the $Q_{total}$ and testing the assumption of homogeneity using a $\chi^2$ distribution [20]. To estimate the magnitude of the true dispersion (real differences among effect sizes), we calculated the $I^2$, which represents the proportion of variance attributable to the between-study variance and not to sampling error [22]. We rejected the assumption of homogeneity and found a high degree of heterogeneity among the effect sizes ($I^2 > 80\%$ for all analyses; electronic supplementary material, table S3).

In an attempt to explain residual heterogeneity on the gender productivity gap, we ran additional analyses including predictor variables (moderators) that could potentially explain effect size variation. In the dataset concerning the gender productivity gap in group-based studies, we performed three different mixed-effects meta-analyses. We tested the effect of the following moderators: (i) research field studied (biological science, social science, health or TEMCP—initials for technology, engineering, mathematic, chemistry and physics), (ii) the century in which the primary data were collected (twentieth or twenty-first century), and (iii) the type of productivity proxy evaluated (peer-reviewed publications, research positions, patents, grants and scientific evaluation committees such as positions in journal editorial boards or grant committees). We examined the $p$-values of $Q_{between}$ statistics, which describe if the variation in effect sizes can be explained by differences among the categories of each moderator. The average of effect sizes was considered significantly different from zero if their 95% confidence intervals did not include the zero value for Hedges' $d$ or the 0.5 value for raw proportion.

## 2.3. Q2: differences in success rate

To investigate differences in success rate among male and female scientists (question 2; electronic supplementary material, dataset S3), we used the natural log of the odds ratio (ln(OR)) [20]. To calculate it, we used the number of successful and failed attempts of male and female scientists. For example, when the productivity proxy was 'articles', the number of successful and failed attempts was the number of published and rejected articles in a scientific journal, respectively. Positive values of ln(OR) indicate that men have higher success rate than women, whereas negative ones indicate higher female success rate.

Data were analysed following the same approach as for Q1. For this dataset, the moderators that we tested were (i) research field and (ii) the type of productivity proxy. Effect sizes were considered significantly different from zero if their 95% confidence intervals did not include the zero value for ln(OR).

## 2.4. Q3: differences in scientific impact

To explore differences in the impact of the science produced by male and female scientists (question 3; electronic supplementary material, dataset S4), we calculated the effect size Hedges' $d$ of studies evaluating quantitative measures of citation numbers, $H$-index or modified versions of this index among genders. All studies used an impact measure that in some way included the citation counts of articles published by men and women. To calculate the Hedges' $d$, we used the mean and the standard deviation of the impact measure and the sample size reported in each study. Positive Hedges' $d$ effect sizes indicate higher impact of male than female scientists, whereas negative ones indicate higher female impact.

Data were analysed following the same approach as for Q1. In this dataset, the moderator that we tested was the inclusion or not of self-citations in the impact measure.

## 2.5. Q4: experimental gender bias

Results from experimental studies that evaluated gender bias in science (question 4; electronic supplementary material, dataset S5) were meta-analysed by calculating the effect size of gender bias as the Hedges' $d$. To obtain this effect size, we used the mean and the standard deviation of the quality rating received by the same piece of science attributed to either a male or a female scientist (according to the name assigned to the author) and the sample size of each study. In a few studies, data were provided as a proportion (e.g. the proportion of the positive and negative reviews of the same work received by male and female authors). For such studies, we first calculated ln(OR), converted it to Cohen's $d$ and finally to the common index Hedges' $d$ to combine all gender bias observations in the same meta-analysis [20]. Positive values of Hedges' $d$ indicate gender bias against women, whereas negative ones indicate gender bias against men. Data were analysed following the same approach than for Q1. Our dataset mostly comprised observations from studies conducted within the Psychology field (61.1%), thus we tested the effect of a moderator that recorded whether or not the study was from the Psychology field.

## 2.6. Publication bias assessment

To address the existence of publication bias and the robustness of the results in each of the five datasets, we used a modification of the Egger's regression appropriate for hierarchical models and sensitivity analyses. We ran Egger's regressions using the residuals of the hierarchical models as the response variable and the effect size precision as the moderator [22]. If the intercept of the Egger's regression was significantly different from zero, this was taken as an evidence of publication bias. Graphical assessment of publication bias (e.g. funnel plots) was not used because its use is still debatable for hierarchical models. We evaluated the sensitivity of all analyses by comparing fitted models with and without effect sizes that were defined as influential outliers. Influential outliers are those extreme values whose exclusion leads to a considerable change in the results of the meta-analysis. Such extreme values were identified using two indicators: hat values and standardized residual values [23]. Effect sizes with hat values greater than two times the average hat value of each dataset and standardized residual values exceeding 3.0 were considered influential outliers [23,24].

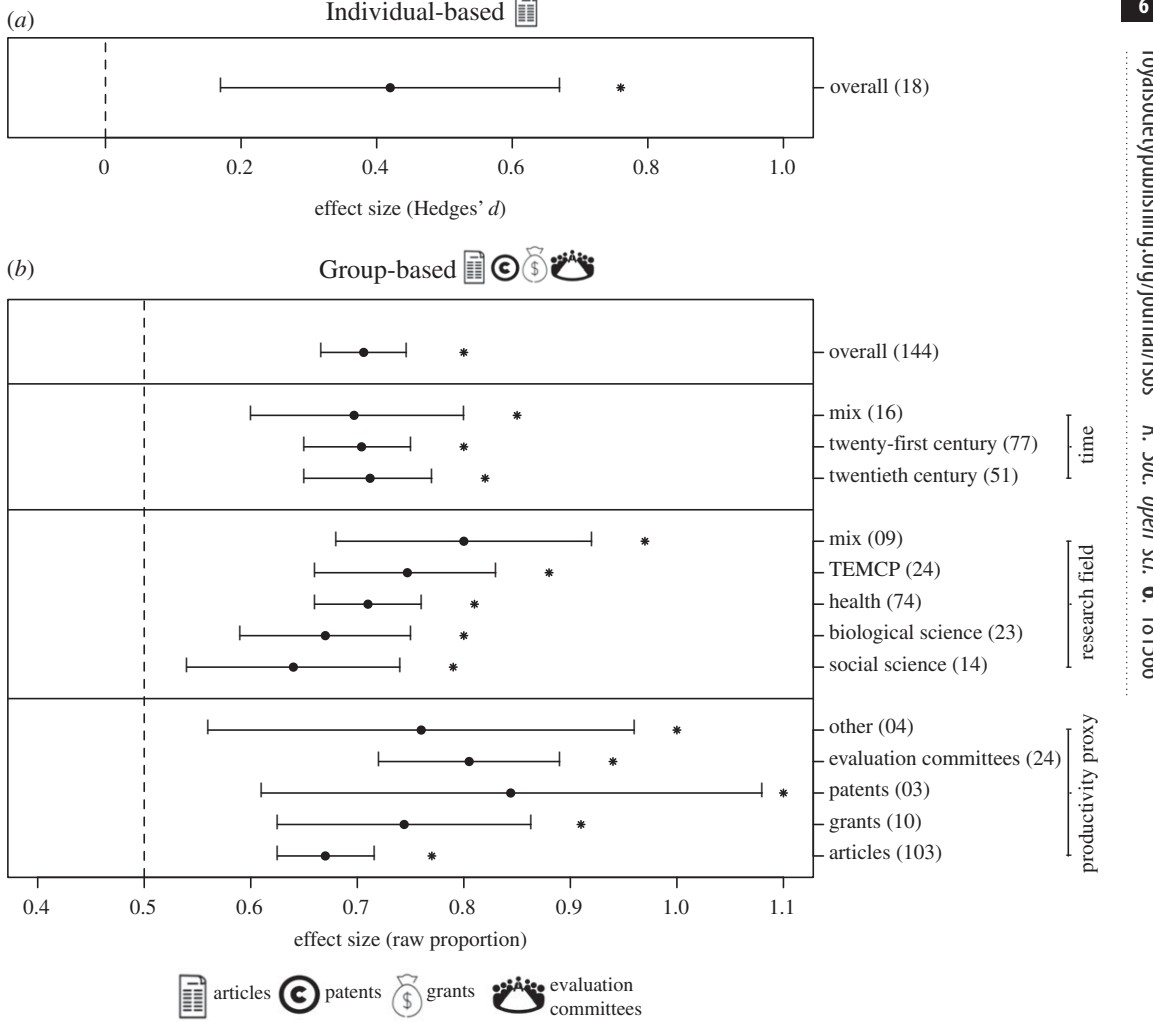

**Figure 2.** The gender productivity gap in science. The mean effect sizes $\pm$ 95% confidence intervals corresponding to (*a*) individual- and (*b*) group-based studies comparing productivity between men and women scientists ($P_{individual} = 0.0012$, $P_{group} < 0.0001$). The number of observations included in each meta-analysis is reported within parentheses. For group-based studies, the effect sizes of gender productivity depending on the productivity proxy ($p = 0.036$), the research field ($p = 0.189$) and the time period ($p = 0.951$) are also shown. The vertical dashed line in each graphic indicates no difference between men and women scientists. Positive effect size values indicate higher men productivity, whereas negative effect sizes indicate higher women productivity. Asterisks denote the mean effect sizes significantly different from zero for Hedges' *d* and 0.5 for raw proportion ($p < 0.05$). Icons illustrate the type of primary response variable included in each meta-analysis.

We used the package metafor (v. 2.0-0) [21] in the R environment (v. 3.5.3, R Core Team, 2019) for all statistical analysis.

# 3. Results

We retrieved 1185 articles from our database search, 110 of which remained suitable for inclusion in meta-analyses (figure 1). From these 110 articles, we calculated 264 effect sizes partitioned into five datasets, according to the question they addressed (electronic supplementary material, datasets S1–S5).

## 3.1. Gender productivity

The meta-analysis of studies investigating mean productivity per capita (i.e. individual-based studies) showed that men published more articles than women (Hedges' $d = 0.418$, number of observations (*o*) = 18, CI = 0.165–0.670; figure 2*a*). Likewise, the meta-analysis of studies comparing the proportion of articles, grants, etc. attributed to men and women (gender-group-based studies) showed that men

produced on average 2.4 times more science than women (raw proportion = 0.706, $o$ = 144, CI = 0.666–0.746; figure 2$b$). Differences in productivity among gender groups did not change in the twenty-first century compared to the twentieth ($p$ = 0.951; twenty-first century: raw proportion = 0.704, $o$ = 77, CI = 0.655–0.754; twentieth century: raw proportion = 0.713, $o$ = 51, CI = 0.654–0.772, figure 2$b$). The gender-group productivity gap was detected in all research fields, being the largest in TEMPC sciences (raw proportion TEMPC = 0.747, $o$ = 24, CI = 0.660–0.833, figure 2$b$), but the differences among areas were not significant ($p$ = 0.189). The productivity of men was higher than that of women for all proxies. However, gender differences varied among proxies ($p$ = 0.036; figure 2$b$). The gender productivity gap measured by group representation in scientific evaluation committees (e.g. research positions, academic evaluations, journal editorial boards) was higher than the gap found for articles production ($p$ = 0.005; evaluation committees: Hedges' $d$ = 0.805, $o$ = 24, CI = 0.718–0.892; articles: Hedges' $d$ = 0.671, $o$ = 103, CI = 0.623–0.716, figure 2$b$).

## 3.2. Gender success rate

Men showed a higher global success rate than women when productivity was weighted by the number of trials (log odds ratio = 0.317, $o$ = 43, CI = 0.141–0.494; figure 3$a$). However, when analysed by research field, we found that men tended to have more success than women only in Health sciences (log odds ratio = 0.419, $o$ = 21, CI = 0.164–0.674; figure 3$a$). Productivity proxy significantly explained part of the variability in success rates among genders ($p$ < 0.001). Men were more successful in gaining faculty or research positions (log odds ratio = 0.368, $o$ = 3, CI = 0.003–0.733), nominations for evaluation committees (log odds ratio = 1.155, $o$ = 2, CI = 0.722–1.587) or grants (log odds ratio = 0.169, $o$ = 27, CI = 0.023–0.315) than expected by their number of trials, but success rate was the same for publishing research articles (log odds ratio = 0.080, $o$ = 8, CI = −0.169–0.327; figure 3$a$).

## 3.3. Gender science impact

The meta-analysis of studies comparing citation numbers, $H$-index or modified versions of this index among genders showed that there is a tendency for men's articles to have globally more impact than those of women (Hedges' $d$ = 0.152, $o$ = 37, CI = −0.008–0.312; $p$ = 0.063; figure 3$b$). Such difference between genders disappeared when we meta-analysed studies in which self-citations were excluded in the measurement of article impact (included: Hedges' $d$ = 0.172, $o$ = 31, CI = −0.0129–0.357, $p$ = 0.07; excluded: Hedges' $d$ = 0.076, $o$ = 6, CI = −0.350–0.508, $p$ = 0.73; figure 3$b$). It is important to highlight that the sample size of the studies with self-citation excluded is reduced, compromising the statistical power of this result.

## 3.4. Experimental gender bias

We quantitatively reviewed results from experimental studies comparing how a CV or a scientific document (i.e. a paper or a conference abstract) attributed to men and women is perceived and evaluated. In such studies, the names of the authors are experimentally changed to assess the effect of gender in their evaluation. We did not find overall differences in the evaluation of the research and academic background of scientists based on gender (Hedges' $d$ = 0.177, $o$ = 18, CI = −0.168–0.522; figure 3$c$). However, we found evidence of experimental gender bias when only studies from areas different to Psychology were considered (other areas: Hedges' $d$ = 0.491, $o$ = 6, CI = 0.221–0.761; Psychology: Hedges' $d$ = 0.104, $o$ = 11, CI = −0.199–0.406, figure 3$c$). The sample size of the studies from areas other than Psychology is reduced, so these results need to be taken with caution.

## 3.5. Publication bias

According to Egger's regressions, there was no sign of publication bias in the datasets (electronic supplementary material, table S4), with the exception of question 1a about per capita differences in scientific productivity ($p$ = 0.051). Additionally, we did not detect influential outliers in any of the datasets (electronic supplementary material, figure S1).

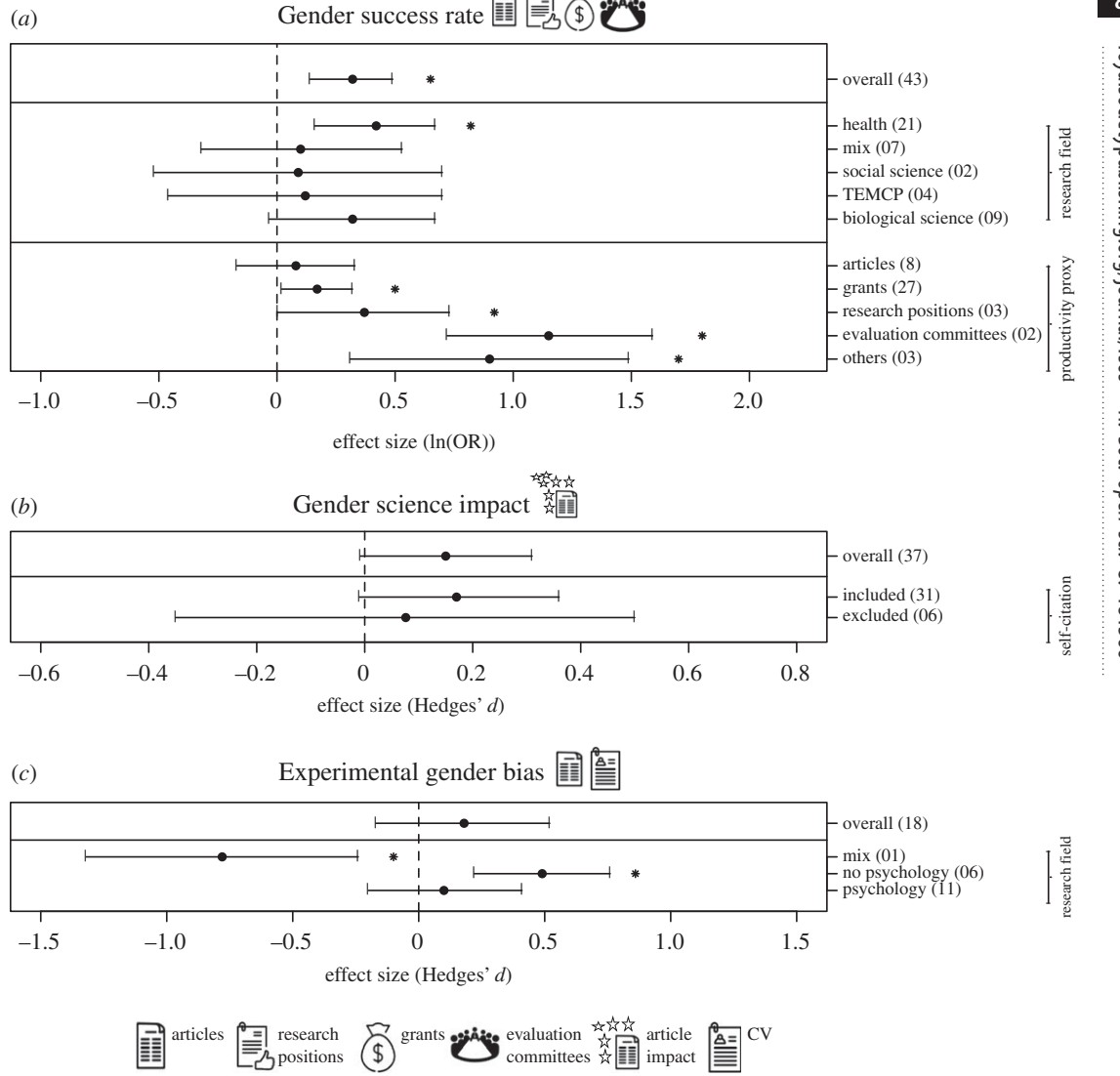

**Figure 3.** Factors that have been associated with the gender productivity gap in science. The mean effect size $\pm$ 95% confidence intervals of (a) gender success rate ($p = 0.0004$), (b) gender science impact ($p = 0.063$) and (c) experimental gender bias ($p = 0.315$). The number of observations included in each meta-analysis is reported within parentheses. The effect size of gender depending on the research field ($p = 0.697$) and the productivity proxy ($p < 0.0001$) are shown for success rate. The effect of self-citations on science impact is also shown ($p = 0.73$). The effect size of gender depending on the research field (Psychology or not, $p = 0.0002$) is shown for experimental gender bias. The vertical dashed line in each graphic indicates no difference between men and women scientists. Effect size values in the right side of each graphic indicate higher men productivity, whereas those in the left side indicate higher women productivity. Asterisks denote the mean effect sizes significantly different from zero ($p < 0.05$). Icons illustrate the type of primary response variables included in each meta-analysis.

## 4. Discussion

### 4.1. The gender productivity gap

Overall, our results support the existence of a gender productivity gap in science. Men published more articles per capita and had more scientific outputs (articles, grants, research positions) as a group. This gender productivity gap has often been used as evidence for meritocracy, being the main cause of the underrepresentation of women in science (i.e. the gender gap). Interestingly, our results showed that most data supporting higher men productivity in science come from gender-group-based studies on productivity (sevenfold more abundant than individual-based ones). This highlights that meritocratic arguments are mainly based on results from group-based studies that do not control the historical overrepresentation of men in the scientific system. If such historical difference is not controlled, it is

rather expected that men as a group produce most of the science, independently of their individual productivity. However, it might be misleading to solely attribute gender differences in productivity to innate differences in scientific abilities between men and women. Many authors have discussed how the development of scientific careers in man-dominated scientific landscapes, particularly at the highest levels of the scientific power structure, may benefit men's careers, mainly by increasing their visualization and rewards [2,7,8,25,26]. Such man-landscape may cause a cumulative disadvantage to the scientific career of women, which has been referred as the 'Matilda effect' [27].

The analysis of the different factors affecting productivity may shed light on the importance of considering the effects of the man-dominated scientific landscape in which men and women develop their scientific careers. Our meta-analysis showed that the size of the gender productivity gap observed in the twentieth century is similar to the one registered in the twenty-first century, which implies that the increase in the representation of women in science in some regions of the world, particularly after the implementation of specific gender policies [2,5,28], has not been enough to deal with the consequences of a historical gender-unbalanced scientific landscape. The marginal differences found in the gender-group productivity gap among research fields may reflect a persistent overrepresentation of men in some research areas [11,29], suggesting that stronger area-specific gender policies are still needed. Last, the overrepresentation of men in evaluation committees may also be the by-product of the 'Matilda effect'. Such overrepresentation can be expected because men are more productive and the selection of scientists for these committees may arise as recognition of productivity. However, the overrepresentation of men as authors of published articles, one of the main proxies of scientists' productivity, was 20% smaller than their overrepresentation in scientific committees. Thus, unravelling how differences in individual productivity between men and women are influenced by the man-dominated scientific landscape in which researchers develop their careers might be of major importance to evaluate the factors producing the gender gap.

## 4.2. Gender success rate

By comparing men's and women's success rates in science, we aimed to explore the most parsimonious explanation underpinning gender productivity differences: the number of trials (i.e. the number of submissions to journals or individuals trying research positions). Interestingly, our meta-analysis on group-based studies showed that both genders have the same success rate in most research fields and when the researcher's work is directly evaluated (e.g. success rate in publishing articles). By contrast, when productivity proxies involve some kind of peer recognition of the researcher's work (e.g. evaluation committees, research positions, grants), men showed higher success rates. These results move forward the discussion on the causes of the gender productivity gap by suggesting that differences in the number of published articles, which is one of the main variables used to evaluate productivity within the scientific community, may be explained by differences in the number of trials. Why do men submit more articles? As discussed before, men as a group may try more because of their overrepresentation in the scientific system, especially at the highest level of academy [2–5]. In top research positions, scientists usually have larger networks of collaborators [25,30] and receive more funding [31], which may allow them to submit more manuscripts and consequently be more productive [32]. On the contrary, women may be disfavoured in the number of trials they can achieve. Moreover, the fact that women have the same success rate in publishing articles as men but do not get research positions, receive grants or are proposed for evaluation committees at the same rate as men may discourage women's scientific careers. These results strongly support the idea that productivity itself may be highly affected by peer recognition and therefore by the scientific landscape in which researchers develop their careers.

Another likely set of explanations of why men may try and produce more is associated with socio-psychological and cultural factors favouring them [7,28]. For instance, it has been reported that men and women dedicate different time to research activities. Women devote more time to teaching or administration [3,33–35], which has been presented as personal choices. However, such choice might be highly influenced by researchers being educated and developing their scientific career in a society that emphasizes gender science stereotypes (i.e. associating research more with men than with women [36]). In this vein, since women have also historically and socially been selected for and have performed domestic and family caring labour, it can be expected that they tend to prioritize family life against work [37–39]. Having less time to do science may affect women's access to funding sources [25,31,40], their ability to make visible their work and to construct a network of collaborators, three key scientific tasks directly related to individual productivity [28,30]. Thus, our results may highlight the importance

of evaluating how socio-psychological and cultural factors may modulate productivity in the scientific system by affecting the number of times that men and women can submit a paper or a grant proposal or even participate in evaluation committees. If these factors still play an important role in determining gender productivity, then gender bias will still affect gender performance [7,8].

## 4.3. Gender science impact

Another explanation for the gender productivity gap that we quantitatively reviewed is that men produce research of higher impact. We found a tendency towards men having globally more scientific impact than women. However, such difference between genders disappeared when impact was computed by excluding self-citation. Self-citation can be the side effect of either men's authority in a given field by constructing on their past work or the consequence of self-promotion guided by self-confidence and mediated by socio-psychological influences [31,41]. Moreover, self-citation rate highly influences the most commonly used citation metric (i.e. the *H*-index [41]). Therefore, by using such index to evaluate researchers' contributions, the scientific community can be rewarding scientific quality but also self-confident behaviour. Men tend to have higher self-citation rates than women, which may explain why their impact increases with time at a higher rate than that of women [41]. Impact metrics are also strongly positively correlated with productivity [41–43]. Such correlation generates a lottery effect [9] where the chance of having more articles with higher citation rates increases among more productive researchers. Citation patterns may also be the result of gender bias against women, as a recent study showed that articles authored by women received 10% fewer citations than expected if the same articles were written by men [44]. If the higher productivity of men is strongly affected by socio-psychological and cultural factors favouring the development of their careers, then by measuring researchers' performance with both productivity and impact, the gender gap will have twofold influence on the productivity gap. Thus, rethinking current ways to assess the contributions of scientists to assign grants, hire and promote them in the scientific system would be critical to shorten the gender productivity gap.

## 4.4. Experimental gender bias

The last and non-excluding explanation for the gender productivity gap is conscious and unconscious gender bias against female scientists. Overall, the evaluation of the research and academic background of scientists did not show bias based on gender. However, when we explicitly separated studies by the research field in which they were carried out, gender bias against women emerged in the group of experiments performed in research fields other than Psychology. The fact that studies in the Psychology field did not show evidence of gender bias may be attributed to the similar representation of both genders in this research field [45]. On the other hand, as differences in the perception about female and male scientists are a broad process affecting many aspects of the scientific life and work [46], the ability of available experimental studies to detect such bias may be constrained because they followed only one strategy to measure gender bias (i.e. formal academic evaluations). Thus, our results highlight the need of both performing more experimental studies measuring gender bias in research fields where women are more underrepresented (e.g. TEMPC sciences) in order to compare them with results in more gender-balanced research fields, and evaluating such bias in a more integrative way by considering the multiple formal and non-formal evaluation processes that researchers face while developing their academic activities.

# 5. Conclusion

Globally, our meta-analyses suggest that the historical underrepresentation of women in science itself and socio-psychological and cultural factors underpinning gender bias against women may modulate gender differences in productivity perpetuating gender inequality in science. Gender-group productivity differences are not decreasing with time, even in research fields in which gender numerical equality has been reached. Thus, much more work needs to be done to exclude gender inequality. If differences in productivity are linked to the time that researchers can dedicate to do science and to peer recognition in a male-dominated landscape, and if science impact has an important component of self-recognition, then socio-cultural gender bias against women may still be a strong factor promoting such inequality. As for other minorities, serious attempts to change women's

underrepresentation in science will need not only to encourage women to enter and persist in the scientific career, but also new policies oriented to build a more egalitarian and heterogeneous scientific community and society. Considering new strategies to assess the quality of individual and group scientific contributions beyond the *mantra* of quantity may certainly help in this sense, as academia obsession with quantity may be killing creativity, reflection and human relationships [47].

Data accessibility. The datasets supporting this article have been uploaded as part of the electronic supplementary material.

Authors' contributions. J.A., E.S.-G. and C.T.C. designed and performed research (i.e. conceived the idea, reviewed the articles and extracted the data included in the meta-analyses); C.T.C. performed the statistical analyses; J.A., E.S.G. and C.T.C. wrote and edited the paper. All authors gave final approval for publication.

Competing interests. We have no competing interests.

Funding. J.A. (grant nos. 2011/09951-2 and 2012/04941-1), E.S.-G. (grant nos. 2011/17968-2 and 2013/02819-7) and C.T.C. (grant no. 2012/09794-7) thank FAPESP-São Paulo Research Foundation for financial support. E.S.-G is funded by the Spanish Ministry of Economy, Industry and Competitivity (IJCI-2015-24947) and by the Generalitat Valenciana (SEJI/2018/024).

Acknowledgements. We thank R. Aguilar, R. Pardini, I. Pérez and F. Hidalgo for comments and insights on previous versions of this manuscript; P. Nieto for insightful discussions about the role of feminisms in constructing a more egalitarian scientific community; E. Santos for discussion on the construction of the statistical models; and J. Hilgard and one anonymous reviewer for their highly valuable comments and discussions. We thank Chameleon Design, D. Hetteix, Lucid Formation, H. Draiman, Artem Kovyazin, and Vectors Market for icons design (www. thenounproject.com). We also want to thank all the researchers who provided us with raw data upon request when they were missing from the original manuscript. J.A. is a researcher of CONICET (Consejo Nacional de Investigaciones Científicas y Técnicas, Argentina).

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
