## [Reviewer comments · Royal Society Open Science]

Review History

RSOS-181566.R0 (Original submission)

Review form: Reviewer 1 (Joseph Hilgard)

Is the manuscript scientifically sound in its present form?

No

Are the interpretations and conclusions justified by the results?

No

Is the language acceptable?

Yes

Is it clear how to access all supporting data?

Yes

Do you have any ethical concerns with this paper?

No

Have you any concerns about statistical analyses in this paper?

Yes

Recommendation?

Major revision is needed (please make suggestions in comments)

Comments to the Author(s)

The authors present a meta-analysis of the purported gender productivity gap and studies examining potential gender bias. This is a relevant and valuable topic. The literature search appears competently done, and I appreciate the use of PRISMA tools. However, I have concerns about how effect sizes were synthesized and inspected for bias.

My main suggestions are as follows: First, you must model the dependency of multiple outcomes within a study, either by averaging them or using some clustering strategy. Second, you should be aware that the number generated by fail-safe N is essentially meaningless. If you want to estimate how bad the bias is and what the bias-adjusted true effect might be, you will need to use an adjustment model like PET-PEESE, p-curve, or selection modeling. Third, I have reservations about adjusting for the number of submissions.

First, I am concerned that multiple outcomes or manipulations from studies were treated as independent. For example, in Figure S2 and database S5, the experiment by Fidell (1970) appears eight times -- one for each male/female candidate comparison. In effect, Fidell (1970) is treated as having 1,176 independent datapoints. Treating outcomes as independent in this fashion risks badly understating the sampling error and giving undue weight to smaller studies with many conditions or outcomes.

A more conservative approach is to average across outcomes within experiments, treating outcomes as perfectly correlated. This would treat Fidell as having 147 independent datapoints. Other approaches, which I am inexperienced in, include multi-level modeling or "robust variance estimation" (Hedges, Tipton, & Johnson, 2010). The R package "robumeta" is available for robust variance estimation.

There may also be value in considering the different qualitative categories of outcome and treating that as a moderator. For example, one may find greater/lesser gender bias in society leadership and lesser/greater gender bias in publication.

Related to this, details in the Datasets_S1-S6.xls datafile are a little sparse. It would be helpful to have a few details regarding which row represents which outcome or contrast from each experiment. Additionally, moderator analysis only seems to have been applied to database S2 and S3, when other analyses also seem to have remarkable and interesting heterogeneity. For example, database S5 reveals gender biases ranging from $g = -5.4$ to $g = 6.7$. Effects of these magnitudes are nearly unheard of in social science -- please double check these!

Second, fail-safe N conveys very little information, and a large fail-safe N does little to ensure a robust effect. In brief, Fail-Safe N can grow very large when there is publication bias or p-hacking. In the case of ego depletion, fail-safe N was 50,000, but no registered report has been able to replicate the effect. See <http://crystalprisonzone.blogspot.com/2016/07/the-failure-of-fail-safe-n.html> for more details.

As an alternative, I would propose that you provide funnel plots for the relevant meta-analyses. You could apply PET-PEESE (Stanley & Doucouliagos, 2014), p-curve (Simonsohn, Nelson, & Simmons, 2014), or selection modeling (Hedges, 1992) to try to estimate a bias-adjusted effect size. Software packages include <https://cran.r-project.org/web/packages/weightr/weightr.pdf>

and <https://github.com/RobbievanAert/puniform> and p-curve.com. You may read about the properties of these adjustments in our preprint at <https://osf.io/preprints/psyarxiv/9h3nu> but please do not feel obligated to cite that document.

Third, I'm not sure it's a good idea to control for number of submissions. Consider Dr. A and Dr. B. Dr. A submitted ten papers and four grants, getting five papers published and two grants funded. Dr. B submitted four papers and two grants, getting two papers published and one grant funded. Both have the same "success rate", but I think it is fair to say that Dr. A is more productive and would be more favorably evaluated in hiring decisions. Adjusting for the "number of trials", then, would seem to be a problem of conditioning on the outcome. It is my understanding that most grants have the same poor chances of being funded, and so the battle is generally one of perseverance and productivity, rather than success rate. I urge caution in how you discuss this finding in the introduction and discussion.

I had some difficulty in understanding the discussion. It is not clear to me why the gender-group-based studies would support the meritocratic view of the gender gap. Given the relative base rates of men and women in science, as you point out, is there any use to these group-based studies? And if individual-productivity studies cannot distinguish between "men publish more because of merit" and "men publish more because of greater access to resources", what does this meta-analysis teach us?

Also, while I was able to access the data and other supplementary files, the reviewer materials suggest that I should also be looking for analysis code, which I'm not sure I can access.

Once these issues are addressed I think this could be a useful quantitative review.

I always sign my reviews,
Joseph Hilgard
Assistant Professor, Psychology
Illinois State University

Hedges, L. V. (1992). Modeling publication selection effects in meta-analysis. *Statistical Science*, 246-255.

Simonsohn, U., Nelson, L. D., & Simmons, J. P. (2014). p-curve and effect size: Correcting for publication bias using only significant results. *Perspectives on Psychological Science*, 9(6), 666-681.

Stanley, T. D., & Doucouliagos, H. (2014). Meta-regression approximations to reduce publication selection bias. *Research Synthesis Methods*, 5(1), 60-78.

Review form: Reviewer 2

Is the manuscript scientifically sound in its present form?

No

Are the interpretations and conclusions justified by the results?

No

Is the language acceptable?

No

Is it clear how to access all supporting data?

No

Do you have any ethical concerns with this paper?

No

Have you any concerns about statistical analyses in this paper?

No

Recommendation?

Major revision is needed (please make suggestions in comments)

Comments to the Author(s)

This is an interesting manuscript, presenting the results of five different meta-analyses on gender differences in: (1-2) research productivity (mostly measured as the number of papers); (3) success rate (e.g. number of submitted/successful grants); (4) researcher impact (e.g. H-index); and (5) evaluations (i.e. experimental tests of gender bias).

While I have quite a few concerns with the way the manuscript is currently presented, I think they could all be addressed with a revision.

Ambiguous methods

More detail needs to be provided in the methods, so that someone else could update the meta-analyses in the future if they wanted to.

I think the PRISMA diagram would be helpful in the main text rather than the supplement, given the complexity of how the data was collected and divided into different datasets.

Lines 64-66: "By scanning the bibliographies of these candidate articles, we added new studies that were not originally detected in this first search in the ISI Web of Knowledge."

Which candidate articles were 'scanned', and what does this mean? Does it just mean one person scrolled through the references and pulled out anything that looked relevant, or was it more systematic?

How were the articles screened for eligibility? This is a very important step in any systematic review, and it's glossed over in the methods. It is also crucial for interpreting the results, because currently readers don't really know what studies are included in the meta-analysis. Lines 67-68: "Such articles were screened through title and summary to check that they included one of the four questions of our study", and Line 71: "The full text of each selected article was revised to check its suitability for our meta-analyses". Was abstract screening software used? What were the criteria for inclusion and exclusion for each of the questions? Was a decision tree used? How many people screened abstracts and full texts? If multiple people were used, were their decisions checked for consistency? And so on.

The PRISMA diagram says that all articles (94) excluded at the full text stage were excluded for the same reason - "because statistics were not reported". I don't believe this - surely some of the articles included at the abstract stage were excluded for other reasons, such as incorrect experimental design?

The details on data extraction are similarly sparse. Who extracted data and how was it extracted? What was extracted, and why? Why was time just coded as either 20th and 21st century, rather than using the year of the study as a continuous covariate? It is helpful to report all variables for which data was collected - e.g. in a metadata table - not just those that are presented in the manuscript.

Were the data divided into different databases during extraction, and how was this decided?

Lines 90-92 simply says "Suitable studies were grouped into different datasets depending on the

question that they allowed to meta-analyse. Some articles were suitable to investigate more than one question thus they were included in more than one dataset." More information is needed. It would also be excellent if the authors could please upload their analysis code.

Reporting of Results

There is not enough information in the results section. The first paragraph, which contains a description of the dataset, only lists the total number of studies and effect sizes, but provides no information on the sample sizes within the 5 datasets. Given that these data are analysed independently, please provide the number of studies and effect sizes in each dataset.

When I was reading the results text I really wanted to know effect size estimates, sample sizes, and confidence intervals. The reader needs to understand how much data is available to test the questions – including data in subgroups, like with the self-citation question. I know this information is in the figures, but it's distracting to go back and forth between the text and figures. I think the p-values are unnecessary, given that significance is given by the confidence intervals (for this reason, the asterisks in the figures could be removed).

Conclusions not supported

The end of the abstract is: "As men and women scientists show equal success rate and impact, our results suggest that women underrepresentation in science itself modulated the gender productivity gap by perpetuating a women-unfriendly academic environment. Thus, new policies not only to encourage women to follow a scientific career but also to build a more egalitarian and heterogeneous scientific community are needed to close the gender gap in science."

This is the key message that is delivered in the paper, but it doesn't follow from the results.

The result that men and women have equal success rates when they submit papers/grants is comforting. But this just adds to the result of no clear evidence of obvious gender discrimination. (An aside that isn't discussed in the manuscript: a-priori I might have expected men's success rate to be lower than women's due to greater confidence, i.e. more optimistic choice of journals).

The other results showed that men had a higher research output, and their research was more impactful. Sure, the studies that excluded self-citations didn't show a difference in impact, but this subset of the data was tiny ($n = 3$ studies and $n = 6$ effect sizes), so it had very low statistical power to detect a gender difference. The abstract, results, and discussion emphasise this result too much, given that it's based on such little data. The results also found no experimental evidence for gender bias.

Dogmatic tone and superficial discussion

Parts of the manuscript read like an opinion piece. For example, the vague call for policies to create a more "egalitarian and heterogeneous scientific community" at the end of the abstract and discussion, and the proclamation that "gender equality is not only a fundamental human right but also a building block of sustainable and more peaceful societies."

I agree with the authors on both these points, but I think a more neutral tone would make the manuscript more credible (especially given my above concerns about dubious interpretation of the results). Instead I think more specific discussion of the issues that are alluded to would bolster the arguments.

For example, lines 257-259 read: "men have more time to dedicate to research activities due to socio-psychological and cultural factors, and therefore achieve a higher individual productivity, as our results showed." I think this point is really important, but it is only discussed in this hand-waving way that sounds like special pleading.

Those 'socio-psychological and cultural factors' deserve a paragraph or two on their own. The

results of this meta-analysis show that men are more productive and have a higher research impact, and there's no clear bias within academia, so what's going on? Well, our current academic system rewards people who have the resources (both time and money) to prioritise work above all else. Here there's an extensive literature on gender differences in domestic labour and caring roles. Are there then specific interventions that have been proposed or trialled to change the reward structure in academia?

Another example in the discussion where doing a deeper dive into a topic would be useful is the idea of the "benefit of the commons" (line 235 and elsewhere). I confess I did not know what this was really referring to, so a more explicit explanation would help readers like me.

Again, I find the work interesting and commendable and think all these major comments can be addressed.

Minor comments

Line 1 and Line 9: "Women underrepresentation" should be "women's"

Line 6: "men articles" should be "men's"

Line 8: "less underrepresented" - a bit hard to read, might flow better as "better represented"

Line 30: "less scientific papers, receive less grants" - should be "fewer"

Line 35: "caused only by individuals' meritocracy" - clunky sentence, could be something like "determined solely by merit"

Line 42: "scientist woman" - "woman scientist" or "scientist who is a woman"

Line 45: "in the academia" - delete "the"

Lines 52-53: "we quantitatively reviewed 11 studies (figure S1) evaluating gender differences in scientific productivity and their likely causes"

Line 54: "differences in gender productivity" should be "gender differences in productivity"

Line 71: "revised" should be "screened" or "reviewed"

Line 87: "we recorded if the reported impact measurements included or not self-citations", should be "we recorded whether or not the reporting impact measurements included self-citations"

Line 90-91: "allowed to meta-analyse", reads strangely to me, would prefer just "addressed"

Line 93: "allowed to answer", again could be "addressed"

Line 98: "the Hedges' d" - needs citation, and could remove "the"

Line 100: "sample size informed for" - I think this was meant to be "information?"

Lines 102-106: "row proportion" confused me. I wondered if it was just a typo for "raw proportion" which is used later (line 151)?

Line 108: "natural log of the odds ratio (ln(OR))" - again, citation needed

Line 166: “metaphor” should be “metafor”, and the R package needs a citation and version number

Line 173: “allowed to meta-analyse” - “addressed”

Line 197: “men articles” - “men’s articles”

Line 202-203: “how research and academic background attributed to men and women are perceived and evaluated” - I found this sentence hard to understand

Line 210: “Meta-analyses publication bias” - redundant “Meta-analyses”

Line 223: “Such gap has” → “This gap has” or “The gender productivity gap has”

Line 230: “have been historically” - delete “been”

Line 238: “21th” - “21st”

Line 241-242: “gender numerical inequity” - strange phrase, how about “over-representation of men”

Line 266: “men articles” → “men’s articles”

Line 294: “fields were women” → “where”

References: This paper isn’t cited, but probably should be:

<https://journals.plos.org/plosbiology/article?id=10.1371/journal.pbio.2004956>

Decision letter (RSOS-181566.R0)

30-Nov-2018

Dear Dr Astegiano,

The editors assigned to your paper ("Unraveling the gender productivity gap in science: a meta-analytical review") have now received comments from reviewers. We would like you to revise your paper in accordance with the referee and Associate Editor suggestions which can be found below (not including confidential reports to the Editor). Please note this decision does not guarantee eventual acceptance.

Please submit a copy of your revised paper before 23-Dec-2018. Please note that the revision deadline will expire at 00.00am on this date. If we do not hear from you within this time then it will be assumed that the paper has been withdrawn. In exceptional circumstances, extensions may be possible if agreed with the Editorial Office in advance. We do not allow multiple rounds of revision so we urge you to make every effort to fully address all of the comments at this stage. If deemed necessary by the Editors, your manuscript will be sent back to one or more of the original reviewers for assessment. If the original reviewers are not available, we may invite new reviewers.

- Data accessibility

If you wish to submit your supporting data or code to Dryad (<http://datadryad.org/>), or modify your current submission to dryad, please use the following link:
<http://datadryad.org/submit?journalID=RSOS&manu=RSOS-181566>

- Competing interests

- Authors' contributions

- Acknowledgements

- Funding statement

Please note that Royal Society Open Science charge article processing charges for all new submissions that are accepted for publication. Charges will also apply to papers transferred to Royal Society Open Science from other Royal Society Publishing journals, as well as papers submitted as part of our collaboration with the Royal Society of Chemistry (<http://rsos.royalsocietypublishing.org/chemistry>). If your manuscript is newly submitted and subsequently accepted for publication, you will be asked to pay the article processing charge, unless you request a waiver and this is approved by Royal Society Publishing. You can find out more about the charges at <http://rsos.royalsocietypublishing.org/page/charges>. Should you have any queries, please contact openscience@royalsociety.org.

on behalf of Professor Carolyn McGettigan (Associate Editor) and Professor Antonia Hamilton (Subject Editor)
openscience@royalsociety.org

Associate Editor's comments (Professor Carolyn McGettigan):

I have now received comments on your manuscript from two experts in the field. You will see that while both reviewers find the work to be potentially publishable, they each have substantial queries covering the methodology, reporting and interpretation of the results, and aspects of the writing. If you choose to revise the paper, I encourage you to address each of the reviewers' points in your response.

Comments to Author:

Reviewers' Comments to Author:

Reviewer: 1

Comments to the Author(s)

The authors present a meta-analysis of the purported gender productivity gap and studies examining potential gender bias. This is a relevant and valuable topic. The literature search

appears competently done, and I appreciate the use of PRISMA tools. However, I have concerns about how effect sizes were synthesized and inspected for bias.

My main suggestions are as follows: First, you must model the dependency of multiple outcomes within a study, either by averaging them or using some clustering strategy. Second, you should be aware that the number generated by fail-safe N is essentially meaningless. If you want to estimate how bad the bias is and what the bias-adjusted true effect might be, you will need to use an adjustment model like PET-PEESE, p-curve, or selection modeling. Third, I have reservations about adjusting for the number of submissions.

First, I am concerned that multiple outcomes or manipulations from studies were treated as independent. For example, in Figure S2 and database S5, the experiment by Fidell (1970) appears eight times -- one for each male/female candidate comparison. In effect, Fidell (1970) is treated as having 1,176 independent datapoints. Treating outcomes as independent in this fashion risks badly understating the sampling error and giving undue weight to smaller studies with many conditions or outcomes.

A more conservative approach is to average across outcomes within experiments, treating outcomes as perfectly correlated. This would treat Fidell as having 147 independent datapoints. Other approaches, which I am inexperienced in, include multi-level modeling or "robust variance estimation" (Hedges, Tipton, & Johnson, 2010). The R package "robumeta" is available for robust variance estimation.

There may also be value in considering the different qualitative categories of outcome and treating that as a moderator. For example, one may find greater/lesser gender bias in society leadership and lesser/greater gender bias in publication.

Related to this, details in the Datasets_S1-S6.xls datafile are a little sparse. It would be helpful to have a few details regarding which row represents which outcome or contrast from each experiment. Additionally, moderator analysis only seems to have been applied to database S2 and S3, when other analyses also seem to have remarkable and interesting heterogeneity. For example, database S5 reveals gender biases ranging from $g = -5.4$ to $g = 6.7$. Effects of these magnitudes are nearly unheard of in social science -- please double check these!

Second, fail-safe N conveys very little information, and a large fail-safe N does little to ensure a robust effect. In brief, Fail-Safe N can grow very large when there is publication bias or p-hacking. In the case of ego depletion, fail-safe N was 50,000, but no registered report has been able to replicate the effect. See <http://crystalprisonzone.blogspot.com/2016/07/the-failure-of-fail-safe-n.html> for more details.

As an alternative, I would propose that you provide funnel plots for the relevant meta-analyses. You could apply PET-PEESE (Stanley & Doucouliagos, 2014), p-curve (Simonsohn, Nelson, & Simmons, 2014), or selection modeling (Hedges, 1992) to try to estimate a bias-adjusted effect size. Software packages include <https://cran.r-project.org/web/packages/weightr/weightr.pdf> and <https://github.com/RobbievanAert/puniform> and p-curve.com. You may read about the properties of these adjustments in our preprint at <https://osf.io/preprints/psyarxiv/9h3nu> but please do not feel obligated to cite that document.

Third, I'm not sure it's a good idea to control for number of submissions. Consider Dr. A and Dr. B. Dr. A submitted ten papers and four grants, getting five papers published and two grants funded. Dr. B submitted four papers and two grants, getting two papers published and one grant funded. Both have the same "success rate", but I think it is fair to say that Dr. A is more productive and would be more favorably evaluated in hiring decisions. Adjusting for the

"number of trials", then, would seem to be a problem of conditioning on the outcome. It is my understanding that most grants have the same poor chances of being funded, and so the battle is generally one of perseverance and productivity, rather than success rate. I urge caution in how you discuss this finding in the introduction and discussion.

I had some difficulty in understanding the discussion. It is not clear to me why the gender-group-based studies would support the meritocratic view of the gender gap. Given the relative base rates of men and women in science, as you point out, is there any use to these group-based studies? And if individual-productivity studies cannot distinguish between "men publish more because of merit" and "men publish more because of greater access to resources", what does this meta-analysis teach us?

Also, while I was able to access the data and other supplementary files, the reviewer materials suggest that I should also be looking for analysis code, which I'm not sure I can access.

Once these issues are addressed I think this could be a useful quantitative review.

I always sign my reviews,
Joseph Hilgard
Assistant Professor, Psychology
Illinois State University

Hedges, L. V. (1992). Modeling publication selection effects in meta-analysis. *Statistical Science*, 246-255.

Simonsohn, U., Nelson, L. D., & Simmons, J. P. (2014). p-curve and effect size: Correcting for publication bias using only significant results. *Perspectives on Psychological Science*, 9(6), 666-681.

Stanley, T. D., & Doucouliagos, H. (2014). Meta-regression approximations to reduce publication selection bias. *Research Synthesis Methods*, 5(1), 60-78.

Reviewer: 2

Comments to the Author(s)

This is an interesting manuscript, presenting the results of five different meta-analyses on gender differences in: (1-2) research productivity (mostly measured as the number of papers); (3) success rate (e.g. number of submitted/successful grants); (4) researcher impact (e.g. H-index); and (5) evaluations (i.e. experimental tests of gender bias).

While I have quite a few concerns with the way the manuscript is currently presented, I think they could all be addressed with a revision.

Ambiguous methods

More detail needs to be provided in the methods, so that someone else could update the meta-analyses in the future if they wanted to.

I think the PRISMA diagram would be helpful in the main text rather than the supplement, given the complexity of how the data was collected and divided into different datasets.

Lines 64-66: "By scanning the bibliographies of these candidate articles, we added new studies that were not originally detected in this first search in the ISI Web of Knowledge."

Which candidate articles were 'scanned', and what does this mean? Does it just mean one person scrolled through the references and pulled out anything that looked relevant, or was it more systematic?

How were the articles screened for eligibility? This is a very important step in any systematic review, and it's glossed over in the methods. It is also crucial for interpreting the results, because currently readers don't really know what studies are included in the meta-analysis. Lines 67-68: "Such articles were screened through title and summary to check that they included one of the four questions of our study", and Line 71: "The full text of each selected article was revised to check its suitability for our meta-analyses". Was abstract screening software used? What were the criteria for inclusion and exclusion for each of the questions? Was a decision tree used? How many people screened abstracts and full texts? If multiple people were used, were their decisions checked for consistency? And so on.

The PRISMA diagram says that all articles (94) excluded at the full text stage were excluded for the same reason - "because statistics were not reported". I don't believe this - surely some of the articles included at the abstract stage were excluded for other reasons, such as incorrect experimental design?

The details on data extraction are similarly sparse. Who extracted data and how was it extracted? What was extracted, and why? Why was time just coded as either 20th and 21st century, rather than using the year of the study as a continuous covariate? It is helpful to report all variables for which data was collected - e.g. in a metadata table - not just those that are presented in the manuscript.

Were the data divided into different databases during extraction, and how was this decided? Lines 90-92 simply says "Suitable studies were grouped into different datasets depending on the question that they allowed to meta-analyse. Some articles were suitable to investigate more than one question thus they were included in more than one dataset." More information is needed. It would also be excellent if the authors could please upload their analysis code.

Reporting of Results

There is not enough information in the results section. The first paragraph, which contains a description of the dataset, only lists the total number of studies and effect sizes, but provides no information on the sample sizes within the 5 datasets. Given that these data are analysed independently, please provide the number of studies and effect sizes in each dataset.

When I was reading the results text I really wanted to know effect size estimates, sample sizes, and confidence intervals. The reader needs to understand how much data is available to test the questions - including data in subgroups, like with the self-citation question. I know this information is in the figures, but it's distracting to go back and forth between the text and figures. I think the p-values are unnecessary, given that significance is given by the confidence intervals (for this reason, the asterisks in the figures could be removed).

Conclusions not supported

The end of the abstract is: "As men and women scientists show equal success rate and impact, our results suggest that women underrepresentation in science itself modulated the gender productivity gap by perpetuating a women-unfriendly academic environment. Thus, new policies not only to encourage women to follow a scientific career but also to build a more egalitarian and heterogeneous scientific community are needed to close the gender gap in science."

This is the key message that is delivered in the paper, but it doesn't follow from the results.

The result that men and women have equal success rates when they submit papers/grants is comforting. But this just adds to the result of no clear evidence of obvious gender discrimination. (An aside that isn't discussed in the manuscript: a-priori I might have expected men's success rate to be lower than women's due to greater confidence, i.e. more optimistic choice of journals).

The other results showed that men had a higher research output, and their research was more impactful. Sure, the studies that excluded self-citations didn't show a difference in impact, but this subset of the data was tiny ($n = 3$ studies and $n = 6$ effect sizes), so it had very low statistical power to detect a gender difference. The abstract, results, and discussion emphasise this result too much, given that it's based on such little data. The results also found no experimental evidence for gender bias.

Dogmatic tone and superficial discussion

Parts of the manuscript read like an opinion piece. For example, the vague call for policies to create a more "egalitarian and heterogeneous scientific community" at the end of the abstract and discussion, and the proclamation that "gender equality is not only a fundamental human right but also a building block of sustainable and more peaceful societies."

I agree with the authors on both these points, but I think a more neutral tone would make the manuscript more credible (especially given my above concerns about dubious interpretation of the results). Instead I think more specific discussion of the issues that are alluded to would bolster the arguments.

For example, lines 257-259 read: "men have more time to dedicate to research activities due to socio-psychological and cultural factors, and therefore achieve a higher individual productivity, as our results showed." I think this point is really important, but it is only discussed in this hand-waving way that sounds like special pleading.

Those 'socio-psychological and cultural factors' deserve a paragraph or two on their own. The results of this meta-analysis show that men are more productive and have a higher research impact, and there's no clear bias within academia, so what's going on? Well, our current academic system rewards people who have the resources (both time and money) to prioritise work above all else. Here there's an extensive literature on gender differences in domestic labour and caring roles. Are there then specific interventions that have been proposed or trialled to change the reward structure in academia?

Another example in the discussion where doing a deeper dive into a topic would be useful is the idea of the "benefit of the commons" (line 235 and elsewhere). I confess I did not know what this was really referring to, so a more explicit explanation would help readers like me.

Again, I find the work interesting and commendable and think all these major comments can be addressed.

Minor comments

Line 1 and Line 9: "Women underrepresentation" should be "women's"

Line 6: "men articles" should be "men's"

Line 8: "less underrepresented" - a bit hard to read, might flow better as "better represented"

Line 30: "less scientific papers, receive less grants" - should be "fewer"

Line 35: "caused only by individuals' meritocracy" - clunky sentence, could be something like "determined solely by merit"

Line 42: "scientist woman" - "woman scientist" or "scientist who is a woman"

Line 45: "in the academia" - delete "the"

Lines 52-53: “we quantitatively reviewed 11 studies (figure S1) evaluating gender differences in scientific productivity and their likely causes”

Line 54: “differences in gender productivity” should be “gender differences in productivity”

Line 71: “revised” should be “screened” or “reviewed”

Line 87: “we recorded if the reported impact measurements included or not self-citations”, should be “we recorded whether or not the reporting impact measurements included self-citations”

Line 90-91: “allowed to meta-analyse”, reads strangely to me, would prefer just “addressed”

Line 93: “allowed to answer”, again could be “addressed”

Line 98: “the Hedges’ d” - needs citation, and could remove “the”

Line 100: “sample size informed for” - I think this was meant to be “information?”

Lines 102-106: “row proportion” confused me. I wondered if it was just a typo for “raw proportion” which is used later (line 151)?

Line 108: “natural log of the odds ratio (ln(OR))” - again, citation needed

Line 166: “metaphor” should be “metafor”, and the R package needs a citation and version number

Line 173: “allowed to meta-analyse” - “addressed”

Line 197: “men articles” - “men’s articles”

Line 202-203: “how research and academic background attributed to men and women are perceived and evaluated” - I found this sentence hard to understand

Line 210: “Meta-analyses publication bias” - redundant “Meta-analyses”

Line 223: “Such gap has” → “This gap has” or “The gender productivity gap has”

Line 230: “have been historically” - delete “been”

Line 238: “21th” - “21st”

Line 241-242: “gender numerical inequity” - strange phrase, how about “over-representation of men”

Line 266: “men articles” → “men’s articles”

Line 294: “fields were women” → “where”

References: This paper isn’t cited, but probably should be:
<https://journals.plos.org/plosbiology/article?id=10.1371/journal.pbio>.

Author's Response to Decision Letter for (RSOS-181566.R0)

See Appendix A.

RSOS-181566.R1 (Revision)

Review form: Reviewer 1 (Joseph Hilgard)

Is the manuscript scientifically sound in its present form?

No

Are the interpretations and conclusions justified by the results?

Yes

Is the language acceptable?

Yes

Is it clear how to access all supporting data?

Yes

Do you have any ethical concerns with this paper?

No

Have you any concerns about statistical analyses in this paper?

Yes

Recommendation?

Major revision is needed (please make suggestions in comments)

Comments to the Author(s)

I thank the authors for the considerable clarifications they have provided in this revision, which have improved my evaluation of the manuscript. However, I have concerns about the accuracy of the effect size extraction that will need to be addressed through double-checking and documentation before I can fully endorse this manuscript.

I wanted to thank the authors for clarifying that they are using multilevel modeling. That addresses my concern about multiple outcomes per study.

I'm struggling to match up the datasets with the supplementary figures. I think the enormous $g = 6.73$ effect size from Knobloch-Westerwick et al. (2013) is an error. Reading that paper, they say "the effect suggested in Hypothesis 1 was found to be significant, $F(1, 226) = 4.52$, $p = .035$, partial $\eta^2 = .020$, because abstracts from male authors ($M_{estimated} = 5.33$, $SE = 0.12$) were associated with greater Scientific Quality than abstracts from female authors ($M_{estimated} = 5.26$, $SE = 0.12$)." None of these statistics are consistent with an effect of $g = 6.73$. Indeed, none of the p-values are smaller than .01, which is rather distressing given that this is a pretty large sample with a within-subjects design -- it should be well-powered.

I wish you had done a little bit more to document the extraction and calculation process. In the case of Knobloch-Westerwick, it's not clear how the effect size was calculated from the mixed-design study. Often a dataset will contain some intermediate columns (means, SDs, ns) and some supporting text (e.g., the direct quote from K-W et al.). In the case that several outcomes were extracted from a single study, it would also be nice to have those labeled. For example, in Steinpreis et al. 1999 which outcomes are coded as which effect sizes? Skimming the paper, I see many potential outcomes: recommendation to hire ($F(1, 124) = 11.34, d = 0.44$), adequate research experience ($F(1, 126) = 8.15, d = 0.37$), recommendation to tenure ($F(1, 102) = .07, d = \sim 0$). Again, I do not see any effect sizes that would seem to match the considerable $d = 2.4$ you have entered.

Here I've just started by inspecting the largest and most implausible effect sizes. I don't know about all the other effect sizes, and I'm not going to be able to inspect them all. But I think you would really benefit from double-checking your effect sizes and providing more detail about how these numbers were obtained.

Other remarkably large effect sizes: Addessi et al. 2012, $g = 1.3, g = 2.8$; Grant et al. $\log OR = -4.75$; Martinez et al. 2015, $g = 2.8, g = 3$. Some of these may be accurate, of course, but they deserve better double-checking and documentation. Additionally, you may want to consider the role of any outliers not just on the overall meta-analytic effect but also on your moderator and publication bias analyses.

Other less important issues:

I appreciate your providing .RMD files for the appendices, but I can't run any of them because I don't have datafiles Q1a.txt, Q1b.txt, ... Q4.txt.

I still dislike the use of fail safe N. The manuscript does little to interpret this statistic, which, again, I think is meaningless. I won't force you, of course, but I think the use of fail safe N is counterproductive. I will point out that the code you use for fail safe N does not appear to use the multilevel model and thus does not account for dependency between effect sizes within studies.

You are correct that there is no good way to adjust for publication bias in the multilevel setting, where studies have multiple outcomes. Still, your Egger regression here has the same form as a PEESE regression, so it may be worth considering. You could similarly perform a PET regression through application of the SE instead of the variance. Alternatively, you could simply average the outcomes and make a funnel plot out of that -- I recognize this is flawed, but sometimes it is useful.

More minor writing issues:

Recurring issues with use of the possessive.

Nonstandard fonts.

Are studies n or k ? I feel that it is more common for k to be the number of studies meta-analysed and n to be the number of observations within studies.

I hope that you can provide these improvements.

I always sign my reviews,

Joe Hilgard

Review form: Reviewer 2

Is the manuscript scientifically sound in its present form?

No

Are the interpretations and conclusions justified by the results?

Yes

Is the language acceptable?

No

Is it clear how to access all supporting data?

Yes

Do you have any ethical concerns with this paper?

No

Have you any concerns about statistical analyses in this paper?

No

Recommendation?

Accept with minor revision (please list in comments)

Comments to the Author(s)

I have read the revised manuscript and the response to reviewer's letter. I appreciate the author's efforts and think the manuscript has been improved, but I have some remaining comments for improvement.

Inclusion/exclusion criteria

The authors have not fully addressed my concerns about the reproducibility of the screening methods. Lines 79-80 read "The full text of each selected article was reviewed to check its suitability for our meta-analyses, i.e. that it explicitly explored at least one of our four questions and provided the statistics required for the meta-analysis." This is still vague, because one can imagine a large range of studies that "address" one of the four questions (as stated in the introduction: "How does productivity vary among male and female scientists? Are productivity differences explained by a different success rate or by the number of trials of each gender? Do men produce higher impact science? Is there a gender bias against women in science that can be evidenced by experimental studies?"). I would like to see explicit and specific inclusion criteria for each question (i.e. the study design and types of data sought for each study)

Keeping track of different questions

I think the methods section would benefit from subheadings so that the four questions are presented separately (as is done for the results section). At the moment the reader has to try and remember what each question is and go back and forth between them, as each part of the methods are presented as "for the first question, for the second question" etc.

Non-independence

The first reviewer raised concerns about how non-independence in effect sizes were dealt with in the dataset. The author's response is that the inclusion of a random effect for study ID accounts for this non-independence. However in order to partition out between-study variance from within-study variance, I think the authors should include an observational-level random effect so that the residual error term is modeled directly (e.g. $\sim 1 \mid \text{obs}$, where $\text{obs} = 1:\text{nrow}(\text{data})$).

The methods could also be clearer on how much non-independence exists in the dataset. Lines 86-90 read “When an article reported outcomes by research field, type of productivity proxy (e.g. number of articles and grants) or academic position (assistant and full professor), we registered such information and considered each outcome as a different observation. When an article reported multiple outcomes across a given time period, each outcome was considered a different observation only when the time difference among outcomes was at least of one decade. When such time difference was smaller we only considered the most recent outcome.” However, when I look at the supplementary data, I see multiple effect sizes from the same studies from the same decade and the same research field (e.g., Nkenke et al. 2015, Franco-Cardenas et al. 2015, Grace et al. 2015, just to name a few). So were data from certain studies excluded for being non-independent, but other data were included? How were these decisions made, and why? And were there other studies of non-independence among effect sizes, such as a shared control, which can cause a covariance among sampling variances? This can be modeled with a covariance matrix in the V argument in metafor.

Sparse information in supplementary data

As the first reviewer mentioned, there’s not much information in the supplementary data, and this hasn’t changed. I also don’t see a justification for excluding the raw data that were used to calculate each effect size? This would allow the data extractions and calculations to be verified by other people, and make the results reproducible (currently, the code produces difference results. The author’s say “The small difference (in the third decimal place) between the following results and the results presented in the manuscript are due to the difference in the precision of the original data (more than 15 decimal places) and the dataset made available (8 decimal places).”, but the difference is greater when you look at confidence intervals).

Choice of effect size

Line 114-115: “The effect size of gender in group-based studies was measured by calculation the raw proportion” – why was the proportion used rather than Hedge’s g (which is used for other questions)? And have the authors considered using the log response ratio instead? This can be calculated using the `escalc` function in metafor, and has better statistical properties than the raw proportion (See Hedges et al. 1999: “The meta-analysis of response ratios in experimental ecology”).

Presentation of results

Because different effect sizes are used for different questions, I think it’s important that the results section specifies what each effect size is, so that it can be interpreted quickly (E.g. “Hedges’ g = ”) rather than simply “Effect size = ”).

Also, when the results for moderator variables are given I think the slope estimate and CI should be presented (rather than the intercepts) (e.g. lines 192-195 and 200-201).

Minor comments

Line 12: “associated to” → “associated with”

Line 111 and 132: “such effect size”  “this effect size”

Line 164: “appropriated”  “appropriate”

Line 169: “bring the overall effect to become trivial”  “make the overall effect trivial”

Line 193 and 194: “21th”  “21st”

Line 242: "high men productivity"  "higher male productivity"

Line 247-248: sentence needs reworking, e.g. "However, it might be misleading to solely attribute gender differences in productivity to innate differences in scientific abilities between men and women"

Decision letter (RSOS-181566.R1)

20-Feb-2019

Dear Dr Astegiano:

Manuscript ID RSOS-181566.R1 entitled "Unravelling the gender productivity gap in science: a meta-analytical review" which you submitted to Royal Society Open Science, has been reviewed. The comments of the reviewer(s) are included at the bottom of this letter.

Please submit a copy of your revised paper before 15-Mar-2019. Please note that the revision deadline will expire at 00.00am on this date. If we do not hear from you within this time then it will be assumed that the paper has been withdrawn. In exceptional circumstances, extensions may be possible if agreed with the Editorial Office in advance.

The journal does not generally allow multiple rounds of revision, and it is exceptional that the Editors have allowed it on this occasion, so we urge you to make every effort to fully address all of the comments at this stage. If deemed necessary by the Editors, your manuscript will be sent back to one or more of the original reviewers for assessment. If the original reviewers are not available we may invite new reviewers.

- Ethics statement

- Data accessibility

- Competing interests

- Authors' contributions

- Acknowledgements

- Funding statement

on behalf of Professor Carolyn McGettigan (Associate Editor) and Antonia Hamilton (Subject Editor)
openscience@royalsociety.org

Associate Editor Comments to Author (Professor Carolyn McGettigan):

Thank you for submitting your revised manuscript. As you can see, both reviewers are pleased with the revisions but they both still have substantial outstanding concerns - as both reviewers are still querying methodological decisions and reporting, I consider the requested revisions to be major. I invite you to address these concerns in one further round of revision, which I intend to send back for a final round of reviewer comments. There is a possibility that the paper will have to be rejected if the reviewers are still not satisfied in this next round, so I urge you to address them as completely as possible in your revisions and response.

Reviewer comments to Author:

Reviewer: 2

Comments to the Author(s)

I have read the revised manuscript and the response to reviewer's letter. I appreciate the author's efforts and think the manuscript has been improved, but I have some remaining comments for improvement.

Inclusion/exclusion criteria

The authors have not fully addressed my concerns about the reproducibility of the screening methods. Lines 79-80 read "The full text of each selected article was reviewed to check its suitability for our meta-analyses, i.e. that it explicitly explored at least one of our four questions and provided the statistics required for the meta-analysis." This is still vague, because one can imagine a large range of studies that "address" one of the four questions (as stated in the introduction: "How does productivity vary among male and female scientists? Are productivity differences explained by a different success rate or by the number of trials of each gender? Do men produce higher impact science? Is there a gender bias against women in science that can be evidenced by experimental studies?"). I would like to see explicit and specific inclusion criteria for each question (i.e. the study design and types of data sought for each study)

Keeping track of different questions

I think the methods section would benefit from subheadings so that the four questions are presented separately (as is done for the results section). At the moment the reader has to try and remember what each question is and go back and forth between them, as each part of the methods are presented as "for the first question, for the second question" etc.

Non-independence

The first reviewer raised concerns about how non-independence in effect sizes were dealt with in the dataset. The author's response is that the inclusion of a random effect for study ID accounts for this non-independence. However in order to partition out between-study variance from within-study variance, I think the authors should include an observational-level random effect so that the residual error term is modeled directly (e.g. $\sim 1 | \text{obs}$, where $\text{obs} = 1:\text{nrow}(\text{data})$).

The methods could also be clearer on how much non-independence exists in the dataset. Lines 86-90 read "When an article reported outcomes by research field, type of productivity proxy (e.g. number of articles and grants) or academic position (assistant and full professor), we registered such information and considered each outcome as a different observation. When an article reported multiple outcomes across a given time period, each outcome was considered a different observation only when the time difference among outcomes was at least of one decade. When such time difference was smaller we only considered the most recent outcome." However, when I

look at the supplementary data, I see multiple effect sizes from the same studies from the same decade and the same research field (e.g., Nkenke et al. 2015, Franco-Cardenas et al. 2015, Grace et al. 2015, just to name a few). So were data from certain studies excluded for being non-independent, but other data were included? How were these decisions made, and why? And were there other studies of non-independence among effect sizes, such as a shared control, which can cause a covariance among sampling variances? This can be modeled with a covariance matrix in the V argument in metafor.

Sparse information in supplementary data

As the first reviewer mentioned, there's not much information in the supplementary data, and this hasn't changed. I also don't see a justification for excluding the raw data that were used to calculate each effect size? This would allow the data extractions and calculations to be verified by other people, and make the results reproducible (currently, the code produces difference results. The author's say "The small difference (in the third decimal place) between the following results and the results presented in the manuscript are due to the difference in the precision of the original data (more than 15 decimal places) and the dataset made available (8 decimal places).", but the difference is greater when you look at confidence intervals).

Choice of effect size

Line 114-115: "The effect size of gender in group-based studies was measured by calculation the raw proportion" - why was the proportion used rather than Hedge's g (which is used for other questions)? And have the authors considered using the log response ratio instead? This can be calculated using the escalc function in metafor, and has better statistical properties than the raw proportion (See Hedges et al. 1999: "The meta-analysis of response ratios in experimental ecology").

Presentation of results

Because different effect sizes are used for different questions, I think it's important that the results section specifies what each effect size is, so that it can be interpreted quickly (E.g. "Hedges' g = ") rather than simply "Effect size = ").

Also, when the results for moderator variables are given I think the slope estimate and CI should be presented (rather than the intercepts) (e.g. lines 192-195 and 200-201).

Minor comments

Line 12: "associated to" → "associated with"

Line 111 and 132: "such effect size"  "this effect size"

Line 164: "appropriated"  "appropriate"

Line 169: "bring the overall effect to become trivial"  "make the overall effect trivial"

Line 193 and 194: "21th"  "21st"

Line 242: "high men productivity"  "higher male productivity"

Line 247-248: sentence needs reworking, e.g. "However, it might be misleading to solely attribute gender differences in productivity to innate differences in scientific abilities between men and women"

Reviewer: 1

Comments to the Author(s)

I thank the authors for the considerable clarifications they have provided in this revision, which have improved my evaluation of the manuscript. However, I have concerns about the accuracy of the effect size extraction that will need to be addressed through double-checking and documentation before I can fully endorse this manuscript.

I wanted to thank the authors for clarifying that they are using multilevel modeling. That addresses my concern about multiple outcomes per study.

I'm struggling to match up the datasets with the supplementary figures. I think the enormous $g = 6.73$ effect size from Knobloch-Westerwick et al. (2013) is an error. Reading that paper, they say "the effect suggested in Hypothesis 1 was found to be significant, $F(1, 226) = 4.52$, $p = .035$, partial $\eta^2 = .020$, because abstracts from male authors ($M_{estimated} = 5.33$, $SE = 0.12$) were associated with greater Scientific Quality than abstracts from female authors ($M_{estimated} = 5.26$, $SE = 0.12$)." None of these statistics are consistent with an effect of $g = 6.73$. Indeed, none of the p -values are smaller than .01, which is rather distressing given that this is a pretty large sample with a within-subjects design -- it should be well-powered.

I wish you had done a little bit more to document the extraction and calculation process. In the case of Knobloch-Westerwick, it's not clear how the effect size was calculated from the mixed-design study. Often a dataset will contain some intermediate columns (means, SDs, ns) and some supporting text (e.g., the direct quote from K-W et al.). In the case that several outcomes were extracted from a single study, it would also be nice to have those labeled. For example, in Steinpreis et al. 1999 which outcomes are coded as which effect sizes? Skimming the paper, I see many potential outcomes: recommendation to hire ($F(1, 124) = 11.34$, $d = 0.44$), adequate research experience ($F(1, 126) = 8.15$, $d = 0.37$), recommendation to tenure ($F(1, 102) = .07$, $d = \sim 0$). Again, I do not see any effect sizes that would seem to match the considerable $d = 2.4$ you have entered.

Here I've just started by inspecting the largest and most implausible effect sizes. I don't know about all the other effect sizes, and I'm not going to be able to inspect them all. But I think you would really benefit from double-checking your effect sizes and providing more detail about how these numbers were obtained.

Other remarkably large effect sizes: Addessi et al. 2012, $g = 1.3$, $g = 2.8$; Grant et al. $\log OR = -4.75$; Martinez et al. 2015, $g = 2.8$, $g = 3$. Some of these may be accurate, of course, but they deserve better double-checking and documentation. Additionally, you may want to consider the role of any outliers not just on the overall meta-analytic effect but also on your moderator and publication bias analyses.

Other less important issues:

I appreciate your providing .RMD files for the appendices, but I can't run any of them because I don't have datafiles Q1a.txt, Q1b.txt, ... Q4.txt.

I still dislike the use of fail safe N . The manuscript does little to interpret this statistic, which, again, I think is meaningless. I won't force you, of course, but I think the use of fail safe N is counterproductive. I will point out that the code you use for fail safe N does not appear to use the multilevel model and thus does not account for dependency between effect sizes within studies.

You are correct that there is no good way to adjust for publication bias in the multilevel setting,

where studies have multiple outcomes. Still, your Egger regression here has the same form as a PEESE regression, so it may be worth considering. You could similarly perform a PET regression through application of the SE instead of the variance. Alternatively, you could simply average the outcomes and make a funnel plot out of that -- I recognize this is flawed, but sometimes it is useful.

More minor writing issues:

Recurring issues with use of the possessive.

Nonstandard fonts.

Are studies n or k ? I feel that it is more common for k to be the number of studies meta-analysed and n to be the number of observations within studies.

I hope that you can provide these improvements.

I always sign my reviews,

Joe Hilgard

Author's Response to Decision Letter for (RSOS-181566.R1)

See Appendix B.

RSOS-181566.R2 (Revision)

Review form: Reviewer 1 (Joseph Hilgard)

Is the manuscript scientifically sound in its present form?

No

Are the interpretations and conclusions justified by the results?

Yes

Is the language acceptable?

Yes

Is it clear how to access all supporting data?

Yes

Do you have any ethical concerns with this paper?

No

Have you any concerns about statistical analyses in this paper?

Yes

Recommendation?

Accept as is

Comments to the Author(s)

I thank the authors for providing some supporting documentation of the effect size extraction. I also thank them for correcting whatever error was involved in the extraction of $d = 6$ from Knobloch-Westerwick.

It is frustrating to note that there are still some errors in the effect size extraction. I also wonder why not provide PET, PEESE, and funnel plots if one is already willing to perform Egger's test.

Remaining effect size extraction errors

I'm still not confident that you've extracted the effect sizes correctly from Steinpress et al. 1999. I'm looking at Figures 5 and 6. I make a chi-squared table using the counts from the figure to see if woman candidates are less likely to be recommended for hiring or tenure compared to a male candidate. The effect sizes I get very closely match those I get from the reported F-values: $d = \sim 0.6$ and ~ 0 , respectively. Your DatasetS5 treats these as means and SDs rather than proportions or a chi-square test. You report effect sizes of $d = 0.99$ and 0.54 , respectively. Please correct me if I am wrong.

I am concerned that a similar confusion of SD and SE may be going on in Martinez et al. 2015. It is hard to tell because the authors do not express whether the error bars denote SDs, SEs, or CIs. I think it is safest to assume that they represent SEs, not SDs, as you have assumed. Maybe you could ask the authors. Also, where are you getting these N/cell numbers?

Regarding Addessi et al., 2012, the reason the effect size is so very large is because the authors have reported the standard error of the mean in Table 2 and you have mistakenly used it as the standard deviation, causing you to overestimate the effect size by a factor of \sqrt{n} . The correct effect sizes are $d = 0.14$ for assistant professors and $d = 0.32$ for full professors.

Given that the errors in the extraction of Steinpress & Addessi were carried forward despite my advice, I am anxious that there may be other errors not caught by the authors' double-check. However, I cannot check all the effect sizes myself, and already feel quite fatigued having had to chase down Steinpress, Addessi, and Martinez.

Regarding publication bias analyses

You've declined to consider my advice regarding bias tests. My understanding is that, regarding Egger's test, the recommendation is to use SE, rather than Variance, as the moderator. See Sterne & Egger (2001). The significance (and hence publishability) of results is a linear function of SE. My skim of your citation 21 doesn't seem to provide an argument for using Variance instead of SE.

The text reads "If the intercept of the Egger's regression was significantly different from zero, this was taken as an evidence of publication bias." I don't think this is accurate. Egger's test considers the slope, not the intercept. In your application, the intercept is the PEESE estimate of the bias-adjusted effect size.

I really do think that if you are going to run an Egger test you might as well supply the funnel plots and PET and PEESE estimates in supplement. I recognize that the funnel plots may be somewhat misleading due to the multilevel shape of the data, but it shows a lot to the reader (e.g., the handful of likely errors in Q1a, the high precision and heterogeneity in Q1b, the possible errors in Q2, the heterogeneity in Q4).

#Minor issues

These are optional things to consider.

Summary: Saying "a higher productivity *ascribed* to men" makes it sound like you are arguing that men are not actually more productive, but only *perceived* as more productive. It would be more effective to say simply "There is a gender productivity gap; men produce more publications, grants, and patents." (depending on whether the individual or the group is what's relevant here)

5.2 Gender success rate

"Interestingly, our meta-analysis on group-based studies showed that both genders have the same success rate in most research fields and in the productivity proxy in which the researcher's work is directly evaluated (e.g. publishing articles)." Doesn't your RQ1 show that men publish more articles than women, even at the individual level? You should be clear that you are talking about the rate of article success, not the total rate of publication.

I think extracting only one outcome per study might impair your ability to test for moderators, e.g. the difference between productivity proxies.

Signed,
-Joseph Hilgard

Sterne, J. A., & Egger, M. (2001). Funnel plots for detecting bias in meta-analysis: guidelines on choice of axis. *Journal of clinical epidemiology*, 54(10), 1046-1055.

Decision letter (RSOS-181566.R2)

07-May-2019

Dear Dr Astegiano:

On behalf of the Editors, I am pleased to inform you that your Manuscript RSOS-181566.R2 entitled "Unravelling the gender productivity gap in science: a meta-analytical review" has been accepted for publication in Royal Society Open Science subject to minor revision in accordance with the referee suggestions. Please find the referees' comments at the end of this email.

The reviewers and Subject Editor have recommended publication, but also suggest some minor revisions to your manuscript. Therefore, I invite you to respond to the comments and revise your manuscript.

- Ethics statement

- Data accessibility

It is a condition of publication that all supporting data are made available either as supplementary information or preferably in a suitable permanent repository. The data accessibility section should state where the article's supporting data can be accessed. This section should also include details, where possible of where to access other relevant research materials such as statistical tools, protocols, software etc can be accessed. If the data has been deposited in

an external repository this section should list the database, accession number and link to the DOI for all data from the article that has been made publicly available. Data sets that have been deposited in an external repository and have a DOI should also be appropriately cited in the manuscript and included in the reference list.

<http://datadryad.org/submit?journalID=RSOS&manu=RSOS-181566.R2>

- **Competing interests**

- **Authors' contributions**

- **Acknowledgements**

- **Funding statement**

Because the schedule for publication is very tight, it is a condition of publication that you submit the revised version of your manuscript before 16-May-2019. Please note that the revision deadline will expire at 00.00am on this date. If you do not think you will be able to meet this date please let me know immediately.

on behalf of Professor Carolyn McGettigan (Associate Editor) and Antonia Hamilton (Subject Editor)
openscience@royalsociety.org

Associate Editor Comments to Author (Professor Carolyn McGettigan):

We have received a final evaluation on your revised manuscript from one of the previous reviewers, and on the basis of this I am recommending that your paper be accepted for publication pending minor revisions. You will see that the reviewer still expresses some concern about the accuracy of the effect size calculations, which warrants further attention. Thus, when submitting your final version of the manuscript, please include a cover letter for the editors in

which you address the discrepancies highlighted by the reviewer. If they have truly identified an error in your method, please perform the necessary corrections in the manuscript and explain in the letter how this has been corrected. If there is no error, please explain why this is the case.

Reviewer comments to Author:

Reviewer: 1

Comments to the Author(s)

I thank the authors for providing some supporting documentation of the effect size extraction. I also thank them for correcting whatever error was involved in the extraction of $d = 6$ from Knoblach-Westerwick.

It is frustrating to note that there are still some errors in the effect size extraction. I also wonder why not provide PET, PEESE, and funnel plots if one is already willing to perform Egger's test.

Remaining effect size extraction errors

I'm still not confident that you've extracted the effect sizes correctly from Steinpress et al. 1999. I'm looking at Figures 5 and 6. I make a chi-squared table using the counts from the figure to see if woman candidates are less likely to be recommended for hiring or tenure compared to a male candidate. The effect sizes I get very closely match those I get from the reported F-values: $d = \sim 0.6$ and ~ 0 , respectively. Your Dataset55 treats these as means and SDs rather than proportions or a chi-square test. You report effect sizes of $d = 0.99$ and 0.54 , respectively. Please correct me if I am wrong.

I am concerned that a similar confusion of SD and SE may be going on in Martinez et al. 2015. It is hard to tell because the authors do not express whether the error bars denote SDs, SEs, or CIs. I think it is safest to assume that they represent SEs, not SDs, as you have assumed. Maybe you could ask the authors. Also, where are you getting these N/cell numbers?

Regarding Addessi et al., 2012, the reason the effect size is so very large is because the authors have reported the standard error of the mean in Table 2 and you have mistakenly used it as the standard deviation, causing you to overestimate the effect size by a factor of \sqrt{n} . The correct effect sizes are $d = 0.14$ for assistant professors and $d = 0.32$ for full professors.

Given that the errors in the extraction of Steinpress & Addessi were carried forward despite my advice, I am anxious that there may be other errors not caught by the authors' double-check. However, I cannot check all the effect sizes myself, and already feel quite fatigued having had to chase down Steinpress, Addessi, and Martinez.

Regarding publication bias analyses

You've declined to consider my advice regarding bias tests. My understanding is that, regarding Egger's test, the recommendation is to use SE, rather than Variance, as the moderator. See Sterne & Egger (2001). The significance (and hence publishability) of results is a linear function of SE. My skim of your citation 21 doesn't seem to provide an argument for using Variance instead of SE.

The text reads "If the intercept of the Egger's regression was significantly different from zero, this was taken as an evidence of publication bias." I don't think this is accurate. Egger's test considers the slope, not the intercept. In your application, the intercept is the PEESE estimate of the bias-adjusted effect size.

I really do think that if you are going to run an Egger test you might as well supply the funnel plots and PET and PEESE estimates in supplement. I recognize that the funnel plots may be somewhat misleading due to the multilevel shape of the data, but it shows a lot to the reader (e.g., the handful of likely errors in Q1a, the high precision and heterogeneity in Q1b, the possible errors in Q2, the heterogeneity in Q4).

#Minor issues

These are optional things to consider.

Summary: Saying "a higher productivity *ascribed* to men" makes it sound like you are arguing that men are not actually more productive, but only *perceived* as more productive. It would be more effective to say simply "There is a gender productivity gap; men produce more publications, grants, and patents." (depending on whether the individual or the group is what's relevant here)

5.2 Gender success rate

"Interestingly, our meta-analysis on group-based studies showed that both genders have the same success rate in most research fields and in the productivity proxy in which the researcher's work is directly evaluated (e.g. publishing articles)." Doesn't your RQ1 show that men publish more articles than women, even at the individual level? You should be clear that you are talking about the rate of article success, not the total rate of publication.

I think extracting only one outcome per study might impair your ability to test for moderators, e.g. the difference between productivity proxies.

Signed,

-Joseph Hilgard

Sterne, J. A., & Egger, M. (2001). Funnel plots for detecting bias in meta-analysis: guidelines on choice of axis. *Journal of clinical epidemiology*, 54(10), 1046-1055.

Author's Response to Decision Letter for (RSOS-181566.R2)

See Appendix C.

Decision letter (RSOS-181566.R3)

23-May-2019

Dear Dr Astegiano,

I am pleased to inform you that your manuscript entitled "Unravelling the gender productivity gap in science: a meta-analytical review" is now accepted for publication in Royal Society Open Science.

on behalf of Professor Carolyn McGettigan (Associate Editor) and Antonia Hamilton (Subject Editor)
openscience@royalsociety.org

Follow Royal Society Publishing on Twitter: [@RSocPublishing](https://twitter.com/RSocPublishing)
Follow Royal Society Publishing on Facebook:
<https://www.facebook.com/RoyalSocietyPublishing.FanPage/>
Read Royal Society Publishing's blog: <https://blogs.royalsociety.org/publishing/>

Appendix A

Response to reviewers letter

Dear Professors Carolyn McGettigan and Antonia Hamilton,

Thank you very much for the review of our ms # RSOS-181566, entitled “Unravelling the gender productivity gap in science: a meta-analytical review” and for giving us the opportunity of revising it. We have revised the manuscript based on the suggestions made by the editor and the reviewers, and below we indicate how we have addressed them. Should you need additional information or explanations, please do not hesitate to contact us again.

We want to thank you and the reviewers for giving us this opportunity to improve the manuscript.

Best regards,

Julia Astegiano, Esther Sebastián-González and Camila Castanho

Associate Editor Comments (Professor Carolyn McGettigan):

I have now received comments on your manuscript from two experts in the field. You will see that while both reviewers find the work to be potentially publishable, they each have substantial queries covering the methodology, reporting and interpretation of the results, and aspects of the writing. If you choose to revise the paper, I encourage you to address each of the reviewers' points in your response.

Response: Thank you very much for the review and for the valuable comments on the manuscript. We have revised all the text according to the reviewers' suggestions and its quality has largely improved. We hope it is now acceptable for publication in Royal Society Open Science.

Reviewer: 1

The authors present a meta-analysis of the purported gender productivity gap and studies examining potential gender bias. This is a relevant and valuable topic. The literature search appears competently done, and I appreciate the use of PRISMA tools. However, I have concerns about how effect sizes were synthesized and inspected for bias.

Response: We are happy Dr. Hilgard was positive on our paper, and on the use of PRISMA tools. We hope we have been able to deal with all his concerns in this review. Below we detail the changes made to address them.

My main suggestions are as follows: First, you must model the dependency of multiple outcomes within a study, either by averaging them or using some clustering strategy. Second, you should be aware that the number generated by fail-safe N is essentially meaningless. If you want to estimate how bad the bias is and what the bias-adjusted true effect might be, you will need to use an adjustment model like PET-PEESE, p-curve, or selection modelling. Third, I have reservations about adjusting for the number of submissions.

Response: Very important issues, we answer them all below.

First, I am concerned that multiple outcomes or manipulations from studies were treated as independent. For example, in Figure S2 and database S5, the experiment by Fidell (1970) appears eight times -- one for each male/female candidate comparison. In effect, Fidell (1970) is treated as having 1,176 independent data points. Treating outcomes as independent in this fashion risks badly understating the sampling error and giving undue weight to smaller studies with many conditions or outcomes.

A more conservative approach is to average across outcomes within experiments, treating outcomes as perfectly correlated. This would treat Fidell as having 147 independent data points. Other approaches, which I am inexpert in, include multi-level modelling or "robust variance estimation" (Hedges, Tipton, & Johnson, 2010). The R package "robumeta" is available for robust variance estimation.

Response: This is indeed a very important issue, as pseudoreplication, or treating substudies as if they were independent, may produce unrealistic results. In our data analysis we deal with this issue by including the paper ID as a random factor in the modelling (multi-level or hierarchical meta-analysis). This approach has been proved largely appropriate for accounting for the non-independence of the data from a same study (Nakagawa & Santos 2012).

There may also be value in considering the different qualitative categories of outcome and treating that as a moderator. For example, one may find greater/lesser gender bias in society leadership and lesser/greater gender bias in publication.

Response: This is a very interesting way of testing the effect of the moderator. However, we understand that this analysis answers the same question than the moderator analysis we are performing in the study. Indeed, the proposed analysis seems a bit more limited than ours, because it categorizes the outcome before analysing it, while we use a continuous (and thus, more detailed) variable.

Related to this, details in the Datasets_S1-S6.xls datafile are a little sparse. It would be helpful to have a few details regarding which row represents which outcome or contrast from each experiment.

Response: We added a meta-data spreadsheet to Datasets (Dataset S6) to explain the information provided in each column of the five datasets (Dataset S1-S5). Each dataset describes the set of articles and observations that we used to answer each question, as described in the legend of each figure.

Additionally, moderator analysis only seems to have been applied to database S2 and S3, when other analyses also seem to have remarkable and interesting heterogeneity. For example, database S5 reveals gender biases ranging from $g = -$

5.4 to $g = 6.7$. Effects of these magnitudes are nearly unheard of in social science -- please double-check these!

Response: As the reviewer points out, moderator analyses were only applied to databases S2, S3 and S4. We did not test for the effect of any moderator in datasets S1 and S5 because of their small sample sizes (20 and 18, respectively). Because of this small number of cases, the possible moderators that we looked at did not reach the minimum number of samples required to perform an analysis ($n = 3$).

Also, we have revised the magnitudes of the effect sizes and found that one of the extreme values was due a typing error along the calculation of effect size. We thank the reviewer for having realized of that! We fixed this error and obtained the new effect size. The magnitude of the summary effect of question 4 changed without changing the qualitative result, i.e. no gender bias was evidenced by experimental studies.

Second, fail-safe N conveys very little information, and a large fail-safe N does little to ensure a robust effect. In brief, Fail-Safe N can grow very large when there is publication bias or p-hacking. In the case of ego depletion, fail-safe N was 50,000, but no registered report has been able to replicate the effect. See <http://crystalprisonzone.blogspot.com/2016/07/the-failure-of-fail-safe-n.html> for more details.

As an alternative, I would propose that you provide funnel plots for the relevant meta-analyses. You could apply PET-PEESE (Stanley & Doucouliagos, 2014), p-curve (Simonsohn, Nelson, & Simmons, 2014), or selection modelling (Hedges, 1992) to try to estimate a bias-adjusted effect size. Software packages include <https://cran.r-project.org/web/packages/weightr/weightr.pdf> and <https://github.com/RobbievanAert/puniform> and p-curve.com. You may read about the properties of these adjustments in our preprint at <https://osf.io/preprints/psyarxiv/9h3nu> but please do not feel obligated to cite that document.

Response: We thank you the suggested analyses and references. Carter et al. 2017 is a very informative paper about the options available to deal with publication bias. However, the publication bias tools tested in this study are applied to a simple random model as a baseline. As far as we know, most of the tools suggested by the reviewer were not appropriate, or have not yet been implemented for hierarchical models (models that deal with the non-independence arising from multiple effect sizes in single studies). The solutions available are scarce and there seems to be no consensus among experts in meta-analysis about this. Some discussions in specialized forums between researchers and Wolfgang Viechtbauer, the author of the metafor package who is an expert in meta-analytic tools, illustrate the limited tools available to evaluate publication bias for hierarchical models (please see <https://stat.ethz.ch/pipermail/r-sig-meta-analysis/2018-July/000939.html> and <https://stats.stackexchange.com/questions/155693/metafor-package-bias-and-sensitivity-diagnostics>). Among the restricted tools available, we found a modification of the original Egger's regression to deal with the non-independence of data (Nakagawa & Santos 2012, *Evolutionary Ecology* 26: 1253:1274). In such modified version, the residuals of the hierarchical model are regressed by the precision of the effect size. It is a solution because in theory, the meta-analytic residuals are independent of each other and free from effects of heterogeneity (Nakagawa & Santos 2012). Therefore, we have substituted the rank correlation, which is more conservative, to this modified version of Egger's regression. We have decided to

keep the fail-safe N analysis because, although it has some limitations, it can still complement the scarce tools available to deal with publication bias in hierarchical models.

A second concern of the reviewer was to estimate a bias-adjusted effect size. However, because we didn't find any evidence of publication bias, this kind of adjustment was not necessary.

Third, I'm not sure it's a good idea to control for number of submissions. Consider Dr. A and Dr. B. Dr. A submitted ten papers and four grants, getting five papers published and two grants funded. Dr. B submitted four papers and two grants, getting two papers published and one grant funded. Both have the same "success rate", but I think it is fair to say that Dr. A is more productive and would be more favourably evaluated in hiring decisions. Adjusting for the "number of trials", then, would seem to be a problem of conditioning on the outcome. It is my understanding that most grants have the same poor chances of being funded, and so the battle is generally one of perseverance and productivity, rather than success rate. I urge caution in how you discuss this finding in the introduction and discussion.

Response: Thank you for your discussion on this important point of our article. We decided to measure men and women success (i.e. controlling productivity by the number of submissions) because this comparison allowed us to explore the most parsimonious explanation underpinning differences in row productivity: the number of trials. The fact that men and women are equally successful, i.e. have the same relative productivity, allowed us to move forward the discussion about the factors affecting row productivity. Why men try more than women? Is it only because they are better in their scientific work (i.e. meritocratic view)? Or is it also associated to the fact that they have historically been more numerous in the scientific system and therefore accumulate privileges as highlighted by several authors? Is it because they dedicate more time to do science? Then, are socio-psychological and cultural factors favouring men determining such time? We rewrote sentences referring to this topic within the abstract, within paragraphs 2 and 3 of the introduction section and sections 5.1. and 5.2. of the discussion to clarify the importance of considering the number of submissions to unravel the gender productivity gap.

I had some difficulty in understanding the discussion. It is not clear to me why the gender-group-based studies would support the meritocratic view of the gender gap. Given the relative base rates of men and women in science, as you point out, is there any use to these group-based studies? And if individual-productivity studies cannot distinguish between "men publish more because of merit" and "men publish more because of greater access to resources", what does this meta-analysis teach us?

Response: We also appreciate very much your comments on this part of the manuscript. We rewrote the first paragraph of section 5.1 in the discussion to clarify ideas associated to the utility of individual- and group-based studies and we discussed them in the context of previous concepts coming from the sociology of science (the "Matilda effect"; Rossintel 1993). Our main point is that such studies should control by differences in gender incidence, because of its influence on productivity results. Finally, as the reviewer indicates, individual-productivity studies cannot distinguish between "men publish more because of merit" and "men publish more because of greater access to resources", that is why we did not finish our

meta-analysis in this point and we tried to tease apart these effects in the following analysis of the study (i.e. meta-analyses addressing success rate, impact and gender bias).

Also, while I was able to access the data and other supplementary files, the reviewer materials suggest that I should also be looking for analysis code, which I'm not sure I can access.

Response: The R code is now available as supplementary materials Appendices S2-S6.

Hedges, L. V. (1992). Modeling publication selection effects in meta-analysis. *Statistical Science*, 246-255.

Simonsohn, U., Nelson, L. D., & Simmons, J. P. (2014). p-curve and effect size: Correcting for publication bias using only significant results. *Perspectives on Psychological Science*, 9(6), 666-681.

Stanley, T. D., & Doucouliagos, H. (2014). Meta-regression approximations to reduce publication selection bias. *Research Synthesis Methods*, 5(1), 60-78.

Reviewer: 2

This is an interesting manuscript, presenting the results of five different meta-analyses on gender differences in: (1-2) research productivity (mostly measured as the number of papers); (3) success rate (e.g. number of submitted/successful grants); (4) researcher impact (e.g. H-index); and (5) evaluations (i.e. experimental tests of gender bias).

While I have quite a few concerns with the way the manuscript is currently presented, I think they could all be addressed with a revision.

Response: We are glad the reviewer found our study interesting and we thank him/her for very valuable comments that we answer below.

Ambiguous methods

More detail needs to be provided in the methods, so that someone else could update the meta-analyses in the future if they wanted to. I think the PRISMA diagram would be helpful in the main text rather than the supplement, given the complexity of how the data was collected and divided into different datasets.

Response: We totally agree and we have now included the PRISMA diagram in the main text as figure 1.

Lines 64-66: "By scanning the bibliographies of these candidate articles, we added new studies that were not originally detected in this first search in the ISI Web of Knowledge."

Which candidate articles were 'scanned', and what does this mean? Does it just mean one person scrolled through the references and pulled out anything that looked relevant, or was it more systematic?

Response: While reading the full texts of all selected articles (i.e. those that were not excluded after reading the title and summary) to see if they fulfilled our requirements, we always read the references. When we read the title of an article that could be of our interest, we looked in our article database to see if it was there and added it if it was not already listed. We have re-written the sentence to clarify this process (you can read the full description of the screening processes between lines 79 and 98).

How were the articles screened for eligibility? This is a very important step in any systematic review, and it's glossed over in the methods. It is also crucial for interpreting the results, because currently readers don't really know what studies are included in the meta-analysis. Lines 67-68: "Such articles were screened through title and summary to check that they included one of the four questions of our study", and Line 71: "The full text of each selected article was revised to check its suitability for our meta-analyses". Was abstract screening software used? What were the criteria for inclusion and exclusion for each of the questions? Was a decision tree used? How many people screened abstracts and full texts? If multiple people were used, were their decisions checked for consistency? And so on. The PRISMA diagram says that all articles (94) excluded at the full text stage were excluded for the same reason – "because statistics were not reported". I don't believe this – surely some of the articles included at the abstract stage were excluded for other reasons, such as incorrect experimental design?

Response: We have expanded the explanation on how the screening was done to improve quality. Now we specify "articles were manually screened through title and summary to check that they included one of the four questions of our study (figure 1). The screening was done by the three authors following a conservative approach, so that only articles that were clearly out of the scope of the study were excluded at this point", lines 80-82. Because we used such a conservative approach for abstract screening, we did not check for consistency or use a decision tree. However, as now stated "the inclusion or not of unclear articles was personally discussed by the three authors", line 89.

The details on data extraction are similarly sparse. Who extracted data and how was it extracted? What was extracted, and why? Why was time just coded as either 20th and 21st century, rather than using the year of the study as a continuous covariate? It is helpful to report all variables for which data was collected – e.g. in a metadata table – not just those that are presented in the manuscript.

Response: We now specify that the three authors extracted the data from the text and the tables of the articles, but that they also used the DataThief when the information was only available in the figures. We also contacted the authors of the articles when the data was not available in the manuscript.

Time was just coded as 21st and 20th centuries because many articles used data from several years for their analyses; thus, it was not possible to assign them a single year to be used as a continuous variable. We tried to group them within decades, but still a large proportion of the studies belonged to several decades. Thus, we decided to do a simpler but more accurate division in centuries.

As requested, we added more information in each dataset. For example, in the case of Time, we added a second column with the specific year(s) when the primary

data was collected. The same was done for Research field. Please see datasets S1-S5.

Were the data divided into different databases during extraction, and how was this decided? Lines 90-92 simply say, "Suitable studies were grouped into different datasets depending on the question that they allowed to meta-analyse. Some articles were suitable to investigate more than one question thus they were included in more than one dataset." More information is needed.

Response: We have modified the sentence to try to be more specific on how the databases were divided and used. Now we state that: "We created one dataset for each of the four study questions (Datasets S1-S5), except for the first question for which we created two datasets (Datasets S1 and S2). During the extraction process, we filled each dataset with the studies that directly addressed the specific question to be analysed. Some articles were suitable to investigate more than one question thus they were included in more than one dataset."

It would also be excellent if the authors could please upload their analysis code.

Response: The R code is now available as supplementary materials Appendices S2-S6.

Reporting of Results

There is not enough information in the results section. The first paragraph, which contains a description of the dataset, only lists the total number of studies and effect sizes, but provides no information on the sample sizes within the 5 datasets. Given that these data are analysed independently, please provide the number of studies and effect sizes in each dataset.

Response: The samples sizes of all meta-analyses are now available in the new PRISMA figure 1, but also in the results section of the manuscript.

When I was reading the results text I really wanted to know effect size estimates, sample sizes, and confidence intervals. The reader needs to understand how much data is available to test the questions – including data in subgroups, like with the self-citation question. I know this information is in the figures, but it's distracting to go back and forth between the text and figures. I think the p-values are unnecessary, given that significance is given by the confidence intervals (for this reason, the asterisks in the figures could be removed).

Response: We have now added the mean effect sizes, sample sizes and the confidence intervals in the main text to facilitate the reading. However, we have decided to maintain the asterisks in the figure because it helps to identify significant effect sizes.

Conclusions not supported

The end of the abstract is: "As men and women scientists show equal success rate and impact, our results suggest that women underrepresentation in science itself modulated the gender productivity gap by perpetuating a women-unfriendly

academic environment. Thus, new policies not only to encourage women to follow a scientific career but also to build a more egalitarian and heterogeneous scientific community are needed to close the gender gap in science.”

This is the key message that is delivered in the paper, but it doesn't follow from the results. The result that men and women have equal success rates when they submit papers/grants is comforting. But this just adds to the result of no clear evidence of obvious gender discrimination. (An aside that isn't discussed in the manuscript: a-priori I might have expected men's success rate to be lower than women's due to greater confidence, i.e. more optimistic choice of journals).

The other results showed that men had a higher research output, and their research was more impactful. Sure, the studies that excluded self-citations didn't show a difference in impact, but this subset of the data was tiny ($n = 3$ studies and $n = 6$ effect sizes), so it had very low statistical power to detect a gender difference. The abstract, results, and discussion emphasise this result too much, given that it's based on such little data. The results also found no experimental evidence for gender bias.

Response: We thanks the reviewer for discussing our conclusions, we really appreciate these comments. We rewrote the end of abstract and the conclusion section trying to tight between the results and the discussion, and reducing the emphasis on the result that research impact was not different between men and women when excluding self-citation. Besides the lack of clear gender bias against women found in experimental studies, we now underline the importance of other factors that have been related to gender bias in previous studies such as the underrepresentation of women and the socio-cultural factors affecting women productivity in science.

Dogmatic tone and superficial discussion

Parts of the manuscript read like an opinion piece. For example, the vague call for policies to create a more “egalitarian and heterogeneous scientific community” at the end of the abstract and discussion, and the proclamation that “gender equality is not only a fundamental human right but also a building block of sustainable and more peaceful societies.”

Response: We are sorry the reviewer found our tone dogmatic and too full of opinions. We removed that sentence. However, we would like to say that this idea is not ours, but that it is in the most recent UNESCO Global Science report.

I agree with the authors on both these points, but I think a more neutral tone would make the manuscript more credible (especially given my above concerns about dubious interpretation of the results). Instead I think more specific discussion of the issues that are alluded to would bolster the arguments.

For example, lines 257-259 read: “men have more time to dedicate to research activities due to socio-psychological and cultural factors, and therefore achieve a higher individual productivity, as our results showed.” I think this point is really important, but it is only discussed in this hand-waving way that sounds like special pleading.

Those ‘socio-psychological and cultural factors’ deserve a paragraph or two on their own. The results of this meta-analysis show that men are more productive and have a higher research impact, and there's no clear bias within academia, so what's going

on? Well, our current academic system rewards people who have the resources (both time and money) to prioritise work above all else. Here there's an extensive literature on gender differences in domestic labour and caring roles. Are there then specific interventions that have been proposed or trialled to change the reward structure in academia?

Response: We really appreciate these comments from the reviewer as they made us think a lot. We followed all his/her suggestions and edited the abstract, the introduction and the discussion of the manuscript in order to have a more neutral tone and strengthened arguments associated to the parts of the discussion that might be more controversial (particularly sections 5.1 and 5.2). We also added one paragraph in which we discuss socio-psychological and cultural factors underpinning the gender productivity gap as suggested by the reviewer. We recognize the importance of these factors in generating differences among men and women in the time they devote to science, but as we did not have focused on them we decided to add just one paragraph to the discussion. Finally, we added a sentence in the conclusion section about the importance of rethinking the way academia assess researchers' contributions.

Another example in the discussion where doing a deeper dive into a topic would be useful is the idea of the "benefit of the commons" (line 235 and elsewhere). I confess I did not know what this was really referring to, so a more explicit explanation would help readers like me.

Response: Thank you for your comment. We edited both parts of the manuscript in which the concept of the benefit of commons was used (sections 5.1 and at the end of section 5.3). We explained the general idea of the benefit of commons without explicitly using the concept. Instead, we decided to use an old concept developed in the field of sociology of science, "the Matilda effect" (Rossiter 1993), which encapsulates the same ideas and is commonly used in the literature.

Minor comments

Line 1 and Line 9: "Women underrepresentation" should be "women's"

Response: Changed, thanks!

Line 6: "men articles" should be "men's"

Response: Also changed.

Line 8: "less underrepresented" – a bit hard to read, might flow better as "better represented"

Response: Indeed sounds better, changed.

Line 30: "less scientific papers, receive less grants" – should be "fewer"

Response: Changed.

Line 35: “caused only by individuals’ meritocracy” – clunky sentence, could be something like “determined solely by merit”

Response: Changed.

Line 42: “scientist woman” – “woman scientist” or “scientist who is a woman”

Response: Changed.

Line 45: “in the academia” – delete “the”

Response: Deleted.

Lines 52-53: “we quantitatively reviewed 11 studies (figure S1) evaluating gender differences in scientific productivity and their likely causes”

Response: We have double checked the text and the figure and the original number (111) is correct, and not the “11” the reviewer had written.

Line 54: “differences in gender productivity” should be “gender differences in productivity”

Response: Changed.

Line 71: “revised” should be “screened” or “reviewed”

Response: Changed to “reviewed”.

Line 87: “we recorded if the reported impact measurements included or not self-citations”, should be “we recorded whether or not the reporting impact measurements included self-citations”

Response: Changed.

Line 90-91: “allowed to meta-analyse”, reads strangely to me, would prefer just “addressed”

Response: Changed.

Line 93: “allowed to answer”, again could be “addressed”

Response: Changed.

Line 98: “the Hedges’ d” – needs citation, and could remove “the”

Response: Reference added and “the” removed.

Line 100: “sample size informed for” – I think this was meant to be “information?”

Response: You are right, now it is correct.

Lines 102-106: “row proportion” confused me. I wondered if it was just a typo for “raw proportion” which is used later (line 151)?

Response: Yes it is and error, thanks! now we only use “raw”.

Line 108: “natural log of the odds ratio ($\ln(\text{OR})$)” – again, citation needed

Response: Citation added.

Line 166: “metaphor” should be “metafor”, and the R package needs a citation and version number

Response: Changed and citation and version added.

Line 173: “allowed to meta-analyse” – “addressed”

Response: Changed.

Line 197: “men articles” – “men’s articles”

Response: Changed.

Line 202-203: “how research and academic background attributed to men and women are perceived and evaluated” – I found this sentence hard to understand

Response: We have changed the sentence to: “how a CV or a scientific document (i.e. a paper or a conference abstract) attributed to men and women are perceived and evaluated”, now lines 238-240).

Line 210: “Meta-analyses publication bias” – redundant “Meta-analyses”

Response: Changed to “Publication bias”

Line 223: “Such gap has” → “This gap has” or “The gender productivity gap has”

Response: Changed.

Line 230: “have been historically” – delete “been”

Response: Changed.

Line 238: “21th” – “21st”

Response: Changed.

Line 241-242: “gender numerical inequity” – strange phrase, how about “over-representation of men”

Response: Changed.

Line 266: “men articles” → “men’s articles”

Response: Changed.

Line 294: “fields were women” → “where”

Response: Changed.

References: This paper isn’t cited, but probably should be: <https://journals.plos.org/plosbiology/article?id=10.1371/journal.pbio.2004956>

Response: Thanks! It is a very interesting paper. We have now cited it in lines XX

Appendix B

Response to reviewers letter

Dear Professors Carolyn McGettigan and Antonia Hamilton,

Thank you very much for the review of our ms # RSOS-181566.R1, entitled “Unravelling the gender productivity gap in science: a meta-analytical review” and for giving us the opportunity of revising it. We have revised the manuscript based on the suggestions made by the editor and the reviewers, and below we indicate how we have addressed them. Should you need additional information or explanations, please do not hesitate to contact us again.

We want to thank you and the reviewers for giving us this opportunity to improve the manuscript.

Best regards,

Julia Astegiano, Esther Sebastián-González and Camila Castanho

Associate Editor Comments (Professor Carolyn McGettigan):

Thank you for submitting your revised manuscript. As you can see, both reviewers are pleased with the revisions but they both still have substantial outstanding concerns - as both reviewers are still querying methodological decisions and reporting, I consider the requested revisions to be major. I invite you to address these concerns in one further round of revision, which I intend to send back for a final round of reviewer comments. There is a possibility that the paper will have to be rejected if the reviewers are still not satisfied in this next round, so I urge you to address them as completely as possible in your revisions and response.

Response: Thank you very much for the review and for the valuable comments. We have carefully read and revised the text according to the reviewers comments, including:

- Double-checking effect sizes in our full dataset.
- Providing two additional supplementary tables and further details on the text to specify our inclusion criteria (Tables S1-S2).
- Providing new versions of the Supplementary Datasets (Datasets S1-S7) which incorporate raw data to calculate the effect size of each observation used in our study. These new Supplementary Datasets also include criteria related to the design

of the study that allowed us to consider independent observations from the same study, i.e. help to clarify how we dealt with data independence besides the statistical models provided.

- Re-analyzing the data including a new observational-level random term to directly model the residual error term.
- Improving the presentation of the results.

Double-checking all effect sizes in the full dataset and re-analysing the data with a additional random term produced some quantitative and qualitative changes in the results that we now discuss. However, the overall message of our article has not changed. The global gender productivity gap detected in group-based studies is maintained (Q1), but now we did not find differences across time (between the 20th and the 21st century) and research fields. With the new analyses men also showed higher success rates which were associated with the research field (i.e. the Health field is the only one that showed significant differences) and with the productivity proxy that was evaluated (i.e. no differences in publishing articles, but men had higher success rates in getting research positions, grants or places in evaluation committees as editorial boards).

Also, discussing our study with colleagues, we got the suggestion of including a 2-class moderator for our last question (i.e. experimental gender bias). The dataset used for this last question was highly heterogeneous, with more than half of the studies coming from a single research field (Psychology). We mentioned this in the first version of our manuscript but we had not provided an explicit analysis. Thus, following suggestions from our colleagues, we now compare the effect of the research area (Psychology vs. other areas) on the experimental detection of gender bias. We found a lack of gender bias in Psychology, but a significant bias in studies from other fields. We think the inclusion of this new moderator gives further light to the problem of gender bias in science, so we have included it in the study.

We hope this new version satisfies the reviewers and you.

Looking forward to your response.

Julia Astegiano, Esther Sebastián-González and Camila de Toledo Castanho

Reviewer 2

I have read the revised manuscript and the response to reviewer's letter. I appreciate the author's efforts and think the manuscript has been improved, but I have some remaining comments for improvement.

Response: Thanks you very much for the new review and for valuable suggestions. We have carefully followed your suggestions and added more detailed inclusion/exclusion criteria in order to clarify the screening methods, further assessed data independence, added extra information to the supplementary material (tables S1 - S2) and improved the methods section and results presentation. We hope you find this new version much improved.

Inclusion/exclusion criteria

The authors have not fully addressed my concerns about the reproducibility of the screening methods. Lines 79-80 read "The full text of each selected article was reviewed to check its suitability for our meta-analyses, i.e. that it explicitly explored at least one of our four questions and provided the statistics required for the meta-analysis." This is still vague, because one can imagine a large range of studies that "address" one of the four questions (as stated in the introduction: "How does productivity vary among male and female scientists? Are productivity differences explained by a different success rate or by the number of trials of each gender? Do men produce higher impact science? Is there a gender bias against women in science that can be evidenced by experimental studies?"). I would like to see explicit and specific inclusion criteria for each question (i.e. the study design and types of data sought for each study).

Response: We have added two new supplementary tables where we specify our inclusion criteria (table S1) and we describe, for each specific question, which data types, response variables, factors of analysis and additional criteria were used to select the studies to be included in our meta-analyses (table S2). Also, we have rephrased the explanation on the main text to add further details on these criteria in lines 75-110 of the Methods section.

Keeping track of different questions

I think the methods section would benefit from subheadings so that the four questions are presented separately (as is done for the results section). At the moment the reader has to try and remember what each question is and go back and forth between them, as each part of the methods are presented as "for the first question, for the second question" etc.

Response: We have re-organized the methods section following our four studied questions and using subheadings as suggested.

Non-independence

The first reviewer raised concerns about how non-independence in effect sizes were dealt with in the dataset. The author's response is that the inclusion of a random effect for study ID accounts for this non-independence. However in order to partition out between-study variance from within-study variance, I think the authors should include an observational-level random effect so that the residual error term is modeled directly (e.g. $\sim 1|obs$, where $obs = 1:nrow(data)$).

Response: Thanks you very much for realizing of this. Besides the study-level random variable that dealt with the lack of independence of observations from the same study, we added an observation-level random term to directly model the residual error term. This addition resulted in some small changes in our results that we are now discussing. Please see the addition of the random term in methods (lines 130-132) and in the R codes in the Supplementary material.

The methods could also be clearer on how much non-independence exists in the dataset. Lines 86-90 read "When an article reported outcomes by research field, type of productivity proxy (e.g. number of articles and grants) or academic position (assistant and full professor), we registered such information and considered each outcome as a different observation. When an article reported multiple outcomes across a given time period, each outcome was considered a different observation only when the time difference among outcomes was at least of one decade. When such time difference was smaller we only considered the most recent outcome." However, when I look at the supplementary data, I see multiple effect sizes from the same studies from the same decade and the same research field (e.g., Nkenke et al. 2015, Franco-Cardenas et al. 2015, Grace et al. 2015, just to name a few). So were data from certain studies excluded for being non-independent, but other data were included? How were these decisions made, and why? And were there other studies of non-independence among effect sizes, such as a shared control, which can cause a covariance among sampling variances? This can be modeled with a covariance matrix in the V argument in `metafor`.

Response: We are sorry for those unclear cases. To clarify the non-independence of our data we have first added an inclusion criteria checklist for different observations coming from a same study (in the new table S1 and also in the text, lines 98-105). Also, in the new datasets located in the Supplementary material we have added a new column called "Within study independence criteria" with information on what factor makes those cases singular. About the existence of shared control we did not identify different observations from a same study that have shared control.

Sparse information in supplementary data

As the first reviewer mentioned, there's not much information in the supplementary data, and this hasn't changed. I also don't see a justification for excluding the raw data that were used to calculate each effect size? This would allow the data extractions and calculations to be verified by other people, and make the results reproducible (currently, the code produces difference results. The author's say "The small difference (in the third decimal place) between the following results and the results presented in the manuscript are due to the difference in the precision of the original data (more than 15 decimal places) and the dataset made available (8 decimal places).", but the difference is greater when you look at confidence intervals).

Response: We are sorry that you considered our supplementary material incomplete. We have made a great effort to include as much information as possible in the new version. First of all, we have included the raw data used to calculate the effect sizes. Also, we have included extra columns to identify the response variable, type of data and the within-study independence criteria used in each outcome. Finally, because we have now added raw data to the tables, any person can make all the calculations, from effect size to meta-analyses, and there should not be differences associated to the decimals.

Choice of effect size

Line 114-115: "The effect size of gender in group-based studies was measured by calculation the raw proportion" – why was the proportion used rather than Hedge's g (which is used for other questions)? And have the authors considered using the log response ratio instead? This can be calculated using the `escalc` function in `metafor`, and has better statistical properties than the raw proportion (See Hedges et al. 1999: "The meta-analysis of response ratios in experimental ecology").

Response: We measured the effect size of the group-based studies using the raw proportion because this is the kind of effect size that can be calculated with the raw data we had for this question (the proportion of productive units by men, proportion of productive units by female and the total number of productive units). With these data, it is not possible to calculate the Hedges' d . Hedges' d can be calculated from `escalc` function from outcome measures from 2 x 2 table data (in `metafor` `measure="OR2DN"`; as we used for the analysis of Question 4 - please see the analysis code we provide as supplementary information). However that was not the case for Question 1b, in which just a dichotomous proportion is provided.

Presentation of results

Because different effect sizes are used for different questions, I think it's important that the results section specifies what each effect size is, so that it can be interpreted quickly (E.g. "Hedges' $g =$ ") rather than simply "Effect size = ").

Response: Really good suggestion! We have specified the effect size metric throughout the results section.

Also, when the results for moderator variables are given I think the slope estimate and CI should be presented (rather than the intercepts) (e.g. lines 192-195 and 200-201).

Response: We agree with the reviewer that presenting the estimate of the slope instead of the intercept is the most correct when analysing regressions. However, in this case, we are not doing a regression analysis, but we are instead comparing the means of several groups (similar logics to an ANOVA). This type of analysis only provides with one estimate for each group, which is the mean value for each group and is the value presented in results. Please note that the "intercept" term that appears in the summary of the models refers to the first level of the moderator.

Minor comments

Line 12: "associated to" → "associated with"

Response: Changed

Line 111 and 132: "such effect size"  "this effect size"

Response: Changed

Line 164: "appropriated"  "appropriate"

Response: Changed

Line 169: "bring the overall effect to become trivial"  "make the overall effect trivial"

Response: We have excluded the Orwin's fail-safe number analysis from the paper, as suggested by Reviewer 1 and thus this comment does not apply any more.

Line 193 and 194: "21th"  "21st"

Response: Changed

Line 242: “high men productivity”  “higher male productivity”

Response: Due to the changes in the review process, this sentence is not included in the manuscript any more.

Line 247-248: sentence needs reworking, e.g. “However, it might be misleading to solely attribute gender differences in productivity to innate differences in scientific abilities between men and women”

Response: Thanks for the suggestion, we have reworded it accordingly.

Reviewer 1: Joe Hilgard

I thank the authors for the considerable clarifications they have provided in this revision, which have improved my evaluation of the manuscript. However, I have concerns about the accuracy of the effect size extraction that will need to be addressed through double-checking and documentation before I can fully endorse this manuscript.

Response: We are glad that you value our effort to improve the quality of the study using your comments. We are sorry that the accuracy of the effect size extraction was not clear in our previous version.

Following your comments, we carefully reviewed all the raw data extractions (by going back to the original articles), eligibility criteria and effect size calculations and indeed we found some errors. We also modified and unified the inclusion criteria for including different observations from the same study. We have recalculated all effect sizes and we also included in the supplementary material the raw data and more details on each of the observations coming from the same study. Double-checking and introducing modifications in the hierarchical models to model directly the residual error term did not substantially change our global results. The only global result that changed was the one related to question 2, but results from moderators show that the logic followed in the discussion can be maintained (see lines 230-240). We rewrote sections 4.1, 4.2 and 4.3 of the results section, modified statistical results in the text and figures accordingly and re-wrote the discussion section from line 284 to line 338, and from 365 to 369. We hope this new double-checked and verifiable version of our manuscript satisfies you.

I wanted to thank the authors for clarifying that they are using multilevel modeling. That addresses my concern about multiple outcomes per study.

Response: Excellent, thanks!

I'm struggling to match up the datasets with the supplementary figures. I think the enormous $g = 6.73$ effect size from Knobloch-Westerwick et al. (2013) is an error. Reading that paper, they say "the effect suggested in Hypothesis 1 was found to be significant, $F(1, 226) = 4.52$, $p = .035$, partial $\eta^2 = .020$, because abstracts from male authors ($M_{estimated} = 5.33$, $SE = 0.12$) were associated with greater Scientific Quality than abstracts from female authors ($M_{estimated} = 5.26$, $SE = 0.12$)." None of these statistics are consistent with an effect of $g = 6.73$. Indeed, none of the p-values are smaller than .01, which is rather distressing given that this is a pretty large sample with a within-subjects design -- it should be well-powered.

Response: As said above, we have double-checked all the raw data extractions and effect sizes calculations. About the Knobloch-Westerwick et al. (2013) case, you are totally right and this was one of the errors we found. In the new Datasets you can see which raw data we used and the associated (now corrected) effect sizes.

I wish you had done a little bit more to document the extraction and calculation process. In the case of Knobloch-Westerwick, it's not clear how the effect size was calculated from the mixed-design study. Often a dataset will contain some intermediate columns (means, SDs, ns) and some supporting text (e.g., the direct quote from K-W et al.). In the case that several outcomes were extracted from a single study, it would also be nice to have those labeled. For example, in Steinpreis et al. 1999 which outcomes are coded as which effect sizes? Skimming the paper, I see many potential outcomes: recommendation to hire ($F(1, 124) = 11.34$, $d = 0.44$), adequate research experience ($F(1, 126) = 8.15$, $d = 0.37$), recommendation to tenure ($F(1, 102) = .07$, $d = \sim 0$). Again, I do not see any effect sizes that would seem to match the considerable $d = 2.4$ you have entered.

Response: As suggested, we have now included raw data (including those intermediate columns you indicate) and new columns with extra information about what each observation represents. When we included several observations from a single study, we explained what were the differences among them in the "Within study independence criteria" column.

Specifically, in the Steinpreis et al. 1999 study, we chose the two most representative observations for the two groups of tests done. We included one observation for the hierability of a young scientist and another one for the tenurability of a tenure scientist. The data used was extracted from figures 5 and 6 in the paper, and not from the tests that you indicate in the review because those tests are multivariate and then the pure effects cannot be directly extracted.

Here I've just started by inspecting the largest and most implausible effect sizes. I don't know about all the other effect sizes, and I'm not going to be able to inspect them all. But I think you would really benefit from double-checking your effect sizes and

providing more detail about how these numbers were obtained. Other remarkably large effect sizes: Addressi et al. 2012, $g = 1.3$, $g = 2.8$; Grant et al. $\log OR = -4.75$; Martinez et al. 2015, $g = 2.8$, $g = 3$. Some of these may be accurate, of course, but they deserve better double-checking and documentation. Additionally, you may want to consider the role of any outliers not just on the overall meta-analytic effect but also on your moderator and publication bias analyses.

Response: As already stated, we have double-checked the data and added extra information to the dataset tables. Furthermore, we did not find any influential data in the five datasets (fig S1).

Other less important issues:

I appreciate your providing .RMD files for the appendices, but I can't run any of them because I don't have datafiles Q1a.txt, Q1b.txt, ... Q4.txt.

Response: We are sorry that we were not clear, but all those datasets correspond to the different spreadsheets in the "Datasets_S1-S6.txt" file. We have now added, as supplementary material, the txt versions of the datasets to be easily used to run the R codes (Analysis codes Q1-Q4).

I still dislike the use of fail safe N. The manuscript does little to interpret this statistic, which, again, I think is meaningless. I won't force you, of course, but I think the use of fail safe N is counterproductive. I will point out that the code you use for fail safe N does not appear to use the multilevel model and thus does not account for dependency between effect sizes within studies.

Response: We have removed the use of Fail safe number from the manuscript as suggested.

You are correct that there is no good way to adjust for publication bias in the multilevel setting, where studies have multiple outcomes. Still, your Egger regression here has the same form as a PEESE regression, so it may be worth considering. You could similarly perform a PET regression through application of the SE instead of the variance. Alternatively, you could simply average the outcomes and make a funnel plot out of that -- I recognize this is flawed, but sometimes it is useful.

Response: We understand that Egger regression is very similar to PET regression (as described by Carter et al. 2017 page 16). Furthermore, the use of SE and variance should produce similar results as they are operational indicators of the inverse of precision. We do not agree that using funnel plots with the average of the observations from the same study is a better practice than Egger's regression to recognize

publication bias. First, the Egger regression considers the multilevel nature of the model, which is not true for the funnel plot as suggested. On the other hand, the Egger's regression represents the formal statistical test for the visual representation illustrated by a funnel plot. For these reasons, we decided to keep Egger's regressions as the method adopted to address the existence of publication bias in our multilevel datasets.

More minor writing issues:

Recurring issues with use of the possessive.

Response: We have reviewed the use of the possessive and fixed some errors. Please, let us know if we missed any.

Nonstandard fonts.

Response: We think there was a problem while the pdf was created, as indeed there are some weird fonts in the references, captions and headings. We hope this is fixed in the new version.

Are studies n or k ? I feel that it is more common for k to be the number of studies meta-analysed and n to be the number of observations within studies.

Response: You are right, k represents the number of studies in meta-analyses, thanks for noting! We have changed the letters throughout the study.

Appendix C

Response to reviewers letter

Dear Professors Carolyn McGettigan and Antonia Hamilton,

Thank you very much for the acceptance of our ms # RSOS-181566.R2, entitled “Unravelling the gender productivity gap in science: a meta-analytical review” and for giving us the opportunity of revising the minor comments suggested by one of the reviewers. We have revised the manuscript based on the suggestions made by the editor and the reviewer, and below we indicate how we have addressed them. Should you need additional information or explanations, please do not hesitate to contact us again.

We want to thank you and the reviewers for giving us multiple opportunities to improve the manuscript and get it published in the Royal Society Open Science.

Best regards,

Julia Astegiano, Esther Sebastián-González and Camila Castanho

Associate Editor Comments (Professor Carolyn McGettigan):

We have received a final evaluation on your revised manuscript from one of the previous reviewers, and on the basis of this I am recommending that your paper be accepted for publication pending minor revisions. You will see that the reviewer still expresses some concern about the accuracy of the effect size calculations, which warrants further attention. Thus, when submitting your final version of the manuscript, please include a cover letter for the editors in which you address the discrepancies highlighted by the reviewer. If they have truly identified an error in your method, please perform the necessary corrections in the manuscript and explain in the letter how this has been corrected. If there is no error, please explain why this is the case.

Response: Thank you very much for the review and for the valuable comments. We have carefully read and revised the text and dataset according to the comments of the reviewer, including:

- Checking the errors indicated by the reviewers, correcting them and reanalyzing data.
- Adding arguments to explain why we respectfully disagree with the advices of the reviewer related to Egger’s regression and funnel plots uses.

We hope this new version satisfies you.

Looking forward to your response.

Julia Astegiano, Esther Sebastián-González and Camila de Toledo Castanho

Reviewer comments to Author:

Reviewer: 1

Comments to the Author(s)

I thank the authors for providing some supporting documentation of the effect size extraction. I also thank them for correcting whatever error was involved in the extraction of $d = 6$ from Knoblach-Westerwick.

It is frustrating to note that there are still some errors in the effect size extraction. I also wonder why not provide PET, PEESE, and funnel plots if one is already willing to perform Egger's test.

Response: Thank you very much for the new review and for valuable suggestions. We really appreciate your dedication to check our large dataset. We apologize for the small errors you still detected in data extraction. We have checked the errors you indicated and re-analyzed the data. The correction of such errors did not change any of our conclusions.

Remaining effect size extraction errors

I'm still not confident that you've extracted the effect sizes correctly from Steinpress et al. 1999. I'm looking at Figures 5 and 6. I make a chi-squared table using the counts from the figure to see if woman candidates are less likely to be recommended for hiring or tenure compared to a male candidate. The effect sizes I get very closely match those I get from the reported F-values: $d = \sim 0.6$ and ~ 0 , respectively. Your DatasetS5 treats these as means and SDs rather than proportions or a chi-square test. You report effect sizes of $d = 0.99$ and 0.54 , respectively. Please correct me if I am wrong.

Response: In the Steinpress study, the authors do not provide the p-value for one of the outcomes and therefore the calculation of the effect size was not straightforward. We thought about one option, which is the one the reviewer describes: we used the mean and the SD to make our calculations. We were careful and converted the SE to a SD before adding that information. To do so, we averaged the mean and the SE given for male and female reviewers (this is the way the data is presented). The reviewer used an alternative approach, which is to use a chi-square table and got different results. We think that averaging the SE may have produced in-chain small errors that at the end changed the final effect size because the calculations made by the reviewer indeed match better the p-values shown in the paper. Thus, we have used his approach and re-analyzed the data with the new effect sizes calculated by him. The changes in the two effect sizes did not change the qualitative result of this analysis.

I am concerned that a similar confusion of SD and SE may be going on in Martinez et al. 2015. It is hard to tell because the authors do not express whether the error bars denote SDs, SEs, or CIs. I think it is safest to assume that they represent SEs, not SDs, as you have assumed. Maybe you could ask the authors. Also, where are you getting these N/cell numbers?

Response: We thank the reviewer for his comment. It is correct that Martinez et al. do not mention which are the statistics that they report in the figure we used in our meta-analysis

(Figure 4). We assumed that those values were SD instead of SE because they seem too high to be standard errors. We have re-contacted the authors but we did not receive a response. As considering that the reported values correspond to SEs did not change our qualitative results and seem too conservative, we have decided to maintain our original decision. N values were provided by the first author in former emails.

Regarding Addessi et al., 2012, the reason the effect size is so very large is because the authors have reported the standard error of the mean in Table 2 and you have mistakenly used it as the standard deviation, causing you to overestimate the effect size by a factor of \sqrt{n} . The correct effect sizes are $d = 0.14$ for assistant professors and $d = 0.32$ for full professors.

Response: Yes, we did a mistake as the reviewer said. Thank you very much for checking this article. We have changed the calculation of the effect size for this article and re-analyzed the data for question 1. These changes did not alter our qualitative results related to question 1. As Dr. Addessi confirmed in a recent email that all statistics informing variability in that article were standard errors, we also reanalyzed results from question 3. We now found that there is a tendency towards men having higher impact than women, and also towards self-citation explaining this higher impact. Thus, considering these changes we rewrote one sentence in the abstract, two sentences in the result section and one sentence of the discussion section. The text almost suffered no modifications because we originally wrote it considering the fact that our statistical power to test this moderator was small.

Given that the errors in the extraction of Steinpress & Addessi were carried forward despite my advice, I am anxious that there may be other errors not caught by the authors' double-check. However, I cannot check all the effect sizes myself, and already feel quite fatigued having had to chase down Steinpress, Addessi, and Martinez.

Response: We apologize and really appreciate your double check with these three articles. We have gone again through all the studies that used a SD/SE in the calculation of their effect sizes (studies used for questions 1, 3 and 4) and we have not found any other error or doubtful outcome.

Regarding publication bias analyses

You've declined to consider my advice regarding bias tests. My understanding is that, regarding Egger's test, the recommendation is to use SE, rather than Variance, as the moderator. See Sterne & Egger (2001). The significance (and hence publishability) of results is a linear function of SE. My skim of your citation 21 doesn't seem to provide an argument for using Variance instead of SE.

Response: As we argued in the previous letter, the use of SE and variance should produce similar results as they are operational indicators of the inverse of precision. In fact, we run all Egger's tests using SE instead of variance and found similar results. Therefore we decided to keep Egger's tests with variance

The text reads "If the intercept of the Egger's regression was significantly different from zero, this was taken as an evidence of publication bias.". In your application, the intercept is the PEESE estimate of the bias-adjusted effect size.

Response: We followed the recommendation of Nakagawa and Santos (2012; *Evolutionary Ecology*, page 1267) to perform our analyses. Dr. Eduardo Santos is a researcher of the University of São Paulo and we have had the opportunity to personally discuss with him about their approach, which we finally decided to adopt.

I really do think that if you are going to run an Egger test you might as well supply the funnel plots and PET and PEESE estimates in supplement. I recognize that the funnel plots may be somewhat misleading due to the multilevel shape of the data, but it shows a lot to the reader (e.g., the handful of likely errors in Q1a, the high precision and heterogeneity in Q1b, the possible errors in Q2, the heterogeneity in Q4).

Response: We appreciate and respect the point of view of the reviewer but we still argue that a funnel plot as conceived by him does not allow considering the structure of the variance we followed in the general analyses. Therefore, funnel plots will not exactly tell us something clearer than the Egger test. Surprisingly, this was recognized by the reviewer in the previous round; he mentioned that this point was a less important issue, so we do not understand why now seems to be a main point or why he does not accept our criteria. We feel the same with his mention about the fact that our reference does not provide an argument about using variance instead of SE. Well, the argument is that they will produce similar results so we do not see why we need to choose another test if it does not imply an improvement.

#Minor issues

These are optional things to consider.

Summary: Saying "a higher productivity *ascribed* to men" makes it sound like you are arguing that men are not actually more productive, but only *perceived* as more productive. It would be more effective to say simply "There is a gender productivity gap; men produce more publications, grants, and patents." (depending on whether the individual or the group is what's relevant here)

Response: Thank you very much for sharing your opinion about this sentence. We wrote "there is a gender productivity gap mostly supported by a larger scientific production ascribed to men", because we wanted to highlight the fact that this gap is supported by a small number of studies at the individual level and that most studies came from group-based ones -i.e. that do not control for the number of men in the scientific community, creating a bias in the perception of gender productivity at the individual level as discussed in the discussion section.

5.2 Gender success rate

"Interestingly, our meta-analysis on group-based studies showed that both genders have the same success rate in most research fields and in the productivity proxy in which the researcher's work is directly evaluated (e.g. publishing articles)." Doesn't your RQ1 show that men publish more articles than women, even at the individual level? You should be clear that you are talking about the rate of article success, not the total rate of publication.

Response: We rewrote the sentence as follows: "Interestingly, our meta-analysis on group-based studies showed that both genders have the same success rate in most research fields and when the researcher's work is directly evaluated (e.g. success rate in publishing articles)".

I think extracting only one outcome per study might impair your ability to test for moderators, e.g. the difference between productivity proxies.

Response: We do not fully understand the reviewer's concern, as we did select different outcomes from a same study when the different outcomes were answering some of our questions (e.g. they were from different fields of study or productivity proxies)". We limited the outcomes selected from the same study to avoid adding duplicate results or to over-inflate our database with related outcomes. By incorporating different outcomes from the same study we also consider that there are other factors that can influence gender effects.